# Searching for Phase-Locked Variations of the Emission-Line Profiles in Binary Be Stars

Anatoly S. Miroshnichenko [1,2,*], Raghav Chari [3], Stephen Danford [1], Peter Prendergast [4], Alicia N. Aarnio [1], Ivan L. Andronov [5], Lidiia L. Chinarova [5], Aidan Lytle [1], Ainash Amantayeva [6], Ilfa A. Gabitova [6], Nadezhda L. Vaidman [2,6], Sayat S. Baktybayev [6] and Serik A. Khokhlov [6]

[1] Department of Physics and Astronomy, University of North Carolina—Greensboro, Greensboro, NC 27402, USA
[2] Fesenkov Astrophysical Institute, Observatory, 23, Almaty 050020, Kazakhstan
[3] Department of Physics and Astronomy, University of Tennessee, Knoxville, TN 37996, USA
[4] Kernersville Observatory, Winston-Salem, NC 27285, USA
[5] Department of Mathematics, Physics and Astronomy, Odesa National Maritime University, Mechnikova St. 34, 65029 Odesa, Ukraine
[6] Faculty of Physics and Technology, Al-Farabi Kazakh National University, Al-Farabi Ave., 71, Almaty 050040, Kazakhstan
* Correspondence: a_mirosh@uncg.edu

**Abstract:** There is growing evidence that many Be stars are parts of binary systems. As the B-type primaries are very fast rotators and their spectral lines may be distorted by the circumstellar material, it is not easy to measure their radial velocity directly from the spectral lines. It has been shown that some Be binaries exhibit peak intensity variations consisting of double-peaked H$\alpha$ lines that are phase-locked with orbital periods. We searched for such variations in the spectra of 12 Be stars, including several known and suspected binaries. Our results include confirmation of the orbital periods in $\nu$ Geminorum, $\epsilon$ Capricorni, $\kappa$ Draconis, 60 Cygni, and V2119 Cygni, its refinement in o Puppis, as well as suggesting hints for binarity in o Aquarii, BK Camelopardalis, and 10 Cassiopeae. Monitoring of the H$\alpha$ line profile variations in $\beta$ Canis Minoris for over the last 10 years gives further support to the existence of a 182.5-day period found earlier in a smaller set of data. A similar but still preliminary period (179.6 days) was found in the H$\alpha$ line profile variations in $\psi$ Persei. It is shown for the first time that $\nu$ Geminorum exhibits phase-locked variations in the H$\alpha$ emission peak intensity ratio and, therefore, is a part of the inner binary in this triple system. Our results show that the mentioned phase-locked peak intensity variations are observed in more Be binary systems than previously known and can be used to search for binarity of Be stars when application of other methods is inconclusive.

**Keywords:** spectroscopy; binary system; emission-line stars; circumstellar matter; variable stars; data analysis

## 1. Introduction

### 1.1. Binarity of Be Stars

Hot stars with emission-line spectra were discovered over 150 years ago by visual spectroscopy. One of them, $\gamma$ Cassiopeae, was described by Secchi [1] as showing "white lines" in its spectrum. Nearly three dozen other bright stars were found to show both bright and dark lines by the end of the 19th century [2]. When spectral classification was introduced, Merrill et al. [3] presented a separate class of Be stars and published a list of 90 hot emission-line stars which contained several O-type stars and B-type supergiants. A few years later, Struve [4] proposed a model of an unstable single star to explain the emission-line appearance in the spectra of these objects.

However, the need to explain rapid rotation of these stars led to the hypothesis that Be stars may be binary systems undergoing mass transfer between the components [5]. This idea was not very popular until the end of the 20th century, as few Be stars were confirmed binaries. While Jaschek and Egret [6] put together a catalog of 1149 stars brighter than 13th visual magnitude classified as Be, Gies [7] listed 11 Be binaries with confirmed orbital periods and 13 candidate binaries. Harmanec [8] compiled a list of nearly 160 binaries, 28 of which were single-lined spectroscopic Be binaries (with two unconfirmed cases).

McSwain and Gies [9] studied Be stars in 48 open clusters and concluded that ∼75% of Be stars may have gained their rapid rotation through binary transfer, while the rest were born fast rotators. Miroshnichenko [10,11] collected information on the 340 brightest Be stars and showed that ∼50% of those brighter than 4th visual magnitude were verified binaries, while the binary fraction progressively decreased toward fainter stars, indicating that a selection effect was at play. The latter studies also suggested that Be stars with strong emission lines most likely were binaries because the theoretically predicted stellar wind strength for single B-type stars was insufficient to supply enough circumstellar material to produce such emission-line spectra (e.g., [12]). At the same time, Be stars with weak emission lines might still be single.

More recently, Klement et al. [13] studied spectral energy distributions in the disk emission wavelength range (infrared to radio) of 57 Be stars and found reduced radio flux level in 26 objects. These authors explained this phenomenon by possible truncation of the outer parts of the circumstellar disks by secondary components and suggested that the majority of Be stars were spun-up through mass transfer.

Responding to the emerging evidence that a certain fraction of Be stars might have been spun-up by mass transfer, Wang et al. [14] studied UV spectra taken by the IUE space telescope in a search for hot subdwarf components (former mass donors). They found a dozen new binary candidates and detected radial velocity (RV) variations in eight of them, but concluded that the hot components were mostly too faint for detection through the UV excess.

A recent overview of the status of the binary hypothesis, including theoretical developments, and binarity problem in the group of Be stars with peculiar X-ray properpties, called $\gamma$ Cas stars, is provided by Nazé et al. [15]. These authors found 6 new binaries in a sample of 16 such objects, thus further increasing the binary fraction among Be stars.

Summarizing the above studies of the nature of Be stars with an emphasis on their binarity, the following conclusions can be made. Although a large fraction of them are most likely binary systems, detection of the secondary components is not an easy task due to their faintness with respect to the primaries (a typical visual brightness ratio is a few magnitudes). Measuring the RV variations in the absorption lines in the primaries' spectra is difficult due to their fast rotation that makes the lines wide and shallow, thus requiring very high signal-to-noise ratios in the continuum ($\geq$200) and a relatively high spectral resolution power ($R \geq 10{,}000$). Measuring RV variations in the emission lines has usually been performed only for the H$\alpha$ line, whose profiles are not always symmetric and are affected by telluric lines, thus making the results depend on the spectrum rectification procedure and the choice of the line profile region in which to measure RV. Other emission lines, including H$\beta$, which is typically the next-strongest hydrogen line after H$\alpha$, are faint in most cases.

Therefore, other methods of binarity detection, such as spectro-astrometry, combined photometric and spectroscopic monitoring, and searching for periodic variations in the peak intensities of the double-peaked emission-line profiles can be important complementary tools in a quest to reveal the true fraction of binary Be stars. In this paper, we explore the latter method of studying the temporal behavior of the Balmer line profiles that have been shown to exhibit phase-locked variations with the orbital period of known binaries detected by other methods.

### 1.2. Phase-Locked Peak Intensity Variations in the Double-Peaked Line Profiles in Be Stars

Be stars typically exhibit double-peaked emission-line profiles that form in orbiting circumstellar gaseous disks around the B-type star. Different kinds of line profiles are observed when the object is viewed nearly pole-on (single-peaked) or the density distribution of the disk material is significantly disturbed (triple-peaked and even quadruple-peaked). The blue-shifted ("violet") to red-shifted ("red") peak intensity ratio measured with respect to the local continuum is typically denoted as "V/R", and it is usually variable. The V/R variations are explained by structures in the Be star disk (e.g., spiral arms [16]).

Several binary Be stars have exhibited V/R variations synchronized with the orbital period that have been explained by tidal interaction between the stellar components in coplanar systems (c.f., [17] for a theoretical consideration). Non-coplanar systems exhibit disk precession and warping (c.f., [18]). Since most Be binaries seem to be coplanar systems, searching for regular V/R variations may be considered a method of binary detection, as fast-rotating Be stars exhibit broad and shallow photospheric lines which may also be contaminated with the disk emission, thus making detection of the absorption lines and measuring their RVs difficult. At the same time, Be stars with strong emission lines due to high disk densities may exhibit V/R variations with much longer periods than orbital ones (e.g., in 48 Librae).

The V/R method has limited applicability, as Be stars tend to lose their disks (and line emission) or change the double-peaked line profiles with more complicated ones (see, e.g., Section 3.1.1). A list of Be binaries with observed phase-locked V/R variations includes the following objects: 4 Her [19], 59 Cyg [20,21], $\kappa$ Dra [22], $\epsilon$ Cap [23], and $\pi$ Aqr [24]. Rivinius et al. [23] also mention less pronounced phase-locked V/R variations in the B2e+sdO binary $\phi$ Persei, where they have been observed along with longer-term variations.

Our observational program, which includes over 40 Be stars with a visual brightness between $V \sim 2.9$ and $\sim 8.0$ mag, is focused on long-term monitoring of various features, such as emission line appearance/disappearance, V/R variations, and absorption-line positional variations. It covers known binaries and suspected ones, as well as those with a currently single status. One of our particular goals is to verify previously published results on binarity and improve our knowledge of them by using different spectroscopic binary diagnostics based on large sets of homogeneous data.

In this paper, we report the results of our analysis of nearly 1800 optical échelle spectra of 12 Be stars taken by our team. The objects were selected on the basis of the following criteria: (1) bright enough ($V \leq 6$ mag) to achieve high signal-to-noise ratios with the Three College Observatory (hereafter TCO) telescope in a wide spectral region, and (2) the presence of double-peaked Balmer lines that are not very strong to avoid longer-term possible periodicities (see above). We intended to study the V/R behavior in more detail than previously, test the V/R variations as an indicator of binarity, and put more constraints on the applicability of this method. Our data on some of the program stars were supplemented by other available data.

A list of Be stars, whose observations are discussed in this paper, is presented in Table 1. Additional spectra were taken from published sources, such as [23], as well as retrieved from several archives, such as the BeSS (Be Star Spectra) database [1] and the Ritter Observatory of the University of Toledo (Toledo, OH, USA).

Our observations and data reduction are described in Section 2, results on individual stars are reported in Section 3, our findings are discussed in Section 4, and a summary of this study is given in Section 5.

## 2. Observations, Data Reduction, and Analysis

### 2.1. The Data

Most spectroscopic observations analysed here have been taken between 2011 and 2023 at TCO, which is located $\sim$12 km south of the city of Graham, Alamance County, NC, USA. TCO has a 0.81 ṁ telescope equipped with a fiber-fed échelle spectrograph from Shelyak Instruments [2]. The spectrograph uses an ATIK-460EX detector (2749 × 2199 pixels,

pixel size 4.54 μm × 4.54 μm) and provides a spectral resolving power of $R \sim 12{,}000$ in a spectral range from 3800 Å to 7900 Å, without gaps between the spectral orders. More information about TCO and our observing program there can be found in [25].

The TCO data were supplemented by observations taken at the Kernersville Observatory (hereafter KO) located near Winston-Salem, NC, USA, between 2016 and 2018. KO (not operational anymore) was set up by one of us (P. P.) and equipped with a 50 cm Planewave telescope and an échelle spectrograph by Shelyak Instruments, with a smaller wavelength range extracted from the spectra (4175–7900 Å). Nearly 100 spectra of the stars reported here were obtained at KO.

Each spectrum typically consists of several individual exposures, which are summed up during the data reduction process that was done using the *echelle* package in IRAF by A.M. The process includes bias subtraction, spectral order separation, and wavelength calibration using spectra of a ThAr lamp. Flat field images were not taken, because the detector has a pixel sensitivity difference of $\leq 1.5\%$ and the flat field lamp does not cover the entire extracted spectral range.

A typical accuracy of the wavelength calibration was $\sim 300$ m s$^{-1}$, except for a period between September 2018 and October 2020, when the TCO spectrograph was slightly de-focused, that resulted in a systematic RV error of 2.8 km s$^{-1}$. The error was found by nightly observing RV standards (e.g., from [26]) and corrected in all studied spectra. Most spectra have a signal-to-noise ratio $\geq 200$ near the H$\gamma$ line, where RV measurements by cross-correlation were performed, and $\geq 150$ near the H$\alpha$ line, whose parameters were measured for all studied objects but one.

Additionally, we used $\sim 200$ BeSS spectra of $\nu$ Gem, EW Lac, o Aqr, $\beta$ CMi, and V2119 Cyg ($R \sim 10{,}000$ to 20,000) as well as $\sim 100$ spectra of $\nu$ Gem and $\beta$ CMi from the Ritter Observatory ($R \sim 26{,}000$). Most of these spectra were taken in the H$\alpha$ line region. These spectra were re-normalized to the local continuum to remove a wavelength-dependent relative intensity trend, which was present in most of them.

*2.2. Measurements and Corrections*

Since telluric lines contaminate the spectral region around H$\alpha$, we attempted to remove them from the spectra to improve the quality of the V/R measurements. Templates of the telluric line spectra from the TCO and KO spectral resolving power were produced using observations of $\zeta$ Oph taken in a range of zenith angles and at various humidity levels by subtracting its photospheric line profiles. A set of the resulting telluric spectra with different line strengths were interpolated to fit those outside of H$\alpha$. The objects' spectra were then divided by the best-fitting telluric template, which took care of the telluric lines that resided within the line profile. This procedure was most important for the spectra taken during the Summer time, when the humidity was the highest. The TCO and KO observations were not conducted at a humidity level of 80% or higher. We found that only a small fraction ($\sim 5\%$) of our spectra was noticeably affected by telluric lines.

We measured the peak intensities in the H$\alpha$ profiles of all our objects except for o Pup, where the H$\beta$ profiles were utilized (see Section 3.1.6). The spectra were normalized to the local continuum and the peak intensities were measured in units of the continuum without subtracting unity (see right panel of Figure 1). The same procedure was applied to both the H$\alpha$ and H$\beta$ lines.

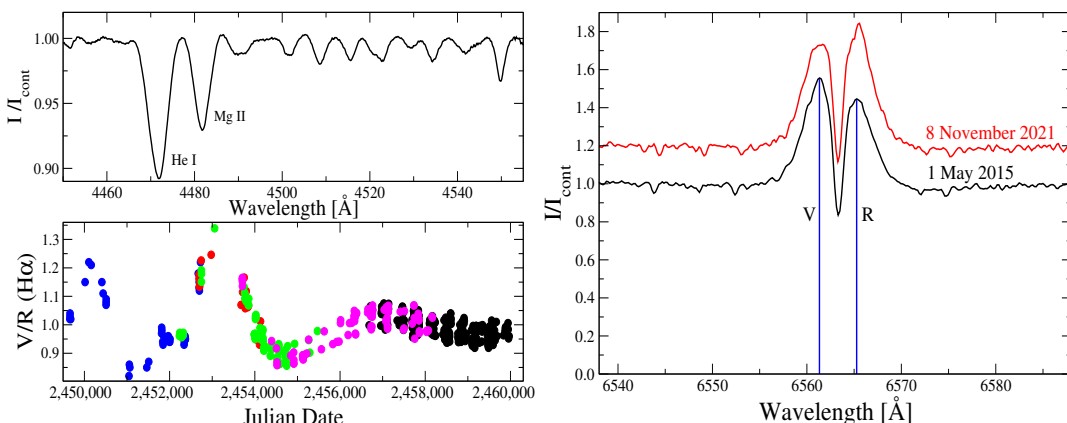

**Figure 1. Left panel**: Top—TCO spectrum of $\nu$ Gem averaged from over 100 individual spectra with a typical signal-to-noise ratio of $\sim$200–300 in the continuum. The spectra were shifted to a common wavelength system before averaging to avoid broadening due to the RV variations. Most of the weak unmarked absorption lines are Fe II lines. Bottom—Temporal behavior of the V/R variations based on published and our data. Symbols: black—TCO and KO, magenta—BeSS, blue—[23], green—Ondřejov, red—RO. Measurement uncertainties are on the order of the symbol size. **Right panel**: Examples of the H$\alpha$ line profiles of $\nu$ Gem with the V/R ratio larger than 1 (black line) and less than 1 (red line). The intensity is normalized to the local continuum; the wavelength scale is heliocentric. The spectra were not corrected for telluric lines. The blue vertical lines show the intensities of the blueshifted (V) and redshifted (R) components of the H$\alpha$ line.

**Table 1.** Be stars observational information.

| Star | V | Sp.T. | Period | JD Range | N |
|------|---|-------|--------|----------|---|
| 1 | 2 | 3 | 4 | 5 | 6 |
| $\nu$ Gem | 4.1 | B6 | 53.73 [23] | 6682–10,025 | 269 |
| $\kappa$ Dra | 3.9 | B6 | 61.55 [27] | 6782–10,071 | 217 |
| V2119 Cyg | 5.9 | B2 | 63.146 [28] | 9396–9823 | 46 |
| $\epsilon$ Cap | 4.6 | B5 | 128.3 [23] | 8004–9845 | 76 |
| 60 Cyg | 5.4 | B1 | 146.6 [29] | 9401–9954 | 29 |
| o Pup | 4.5 | B2 | 28.9 [30] | 7737–10,025 | 65 |
| $\beta$ CMi | 2.9 | B8 | 182.8 $\star$ [31] | 6625–10075 | 290 |
| BK Cam | 4.9 | B2.5 | $-$ [32,33] | 8150–9979 | 98 |
| $\psi$ Per | 4.2 | B5 | $-$ [34] | 6571–9836 | 250 |
| o Aqr | 4.7 | B5 | $-$ [34] | 9049–9946 | 157 |
| EW Lac | 5.4 | B2 | $-$ [35] | 6531–9877 | 205 |
| 10 Cas | 5.6 | B9 | $-$ [35] | 6955–9857 | 105 |

Column information: 1—the object's ID, 2—average brightness of the entire system in the *V*-band, 3—MK type of the Be star, 4—orbital period in days (where available) and references to the papers with information on binarity, 5—Julian date range of the TCO and KO observations (JD–2,450,000), 6—number of spectra taken at TCO and KO. If no orbital period has been published ($-$ in column 4), references to the papers which confirm no secondary companion detection by interferometry are given. $\star$—A similar value for the orbital period (170.4 days) was found in [36] (see discussion in Section 3.2.2).

Sets of absorption lines in a region between 4370 Å and 4600 Å were used for cross-correlation to search for RV variations in the Be star in the spectra of all the objects, but the results turned out to be meaningful only for those with the strongest and narrowest lines (e.g., for $\nu$ Gem, $\kappa$ Dra, and $\epsilon$ Cap). Equivalent widths and bisector RVs of the H$\alpha$ line were measured in a few cases to complement the analysis, such as for o Aqr whose long-term variations have not been summarized in any previous publication. Positions of the central absorptions of the Balmer line profiles were measured in the spectra of $\epsilon$ Cap (from H$\alpha$ to H$\delta$), $\beta$ CMi, $\psi$ Per, o Aqr, and EW Lac (from H$\beta$ to H$\delta$). In our data analysis, we also used measurements of the H$\alpha$ line parameters published for some of our objects (e.g., $\nu$ Gem, EW Lac) as electronic tables (e.g., in [23]).

### 2.3. Time Series Analysis

We have used various methods to analyze our time series. To determine the power spectrum, the software Period04 [37] was used, as well as the online Lomb–Scargle periodogram tool at the NASA Exoplanet Archive site[3]. Hereafter, we call this "the Fourier spectrum", although there are at least 6 modifications of the Fourier Transform (FT) for the (generally) irregularly spaced data with different weights (heteroscedastic data)—see [38] for a review. The most significant peak in the Fourier spectrum was used as an initial approximation to find the best orbital solution. For a more accurate determination of the parameters and their errors, the software MCV[4], based on the algorithms described in [38,39], was used. In cases of noisy phase curves, we only report the most probable period of the V/R variations and show a graph with the data folded with this period.

The phase curve zero-point (T0) in our solutions typically corresponds to a superior conjunction of the Be star for circular orbits (except when described otherwise, e.g., taken from elsewhere) and to a periastron epoch for elliptical orbits (which was only the case for $\nu$ Gem, whose orbital eccentricity was very small and periastron positions were very close to superior conjunctions). If only Fourier analysis was carried out on the V/R data, the zero-point was set to the first observing date.

We have tested the phase curves with a determined period for the presence of eccentricity. Trigonometric polynomials (TP) of the order 1 (circular orbit) and 2 (possible elliptic orbit) were applied to fit the data.

The Fourier coefficients depend on the semi-major axis $a$ of the orbit, the orbital inclination $i$, its eccentricity $e$, and argument of periastron $\omega$ [40]. If the amplitude of a harmonic was statistically significant, we tested the data with a conventional "elliptical" model:

$$V_r = \gamma + K \cdot (\cos(v + \omega) + e \cdot \cos \omega), \tag{1}$$

where $V_r$ is the RV, $K$ is the RV semi-amplitude, $v$ is the true anomaly, $e$ is the orbital eccentricity, and $\omega-$ is the argument of periastron.

In the case of a circular orbit $e = 0$, Equation (1) can be simplified:

$$
\begin{aligned}
V_r &= \gamma + K \cdot (\cos(v + \omega)) = \gamma + K \cdot (\cos v \cos \omega - \sin v \sin \omega) & (2)\\
&= C_1 + C_2 \cdot \cos C_4(t - T_m) + C_3 \cdot \sin C_4(t - T_m) & (3)\\
&= C_1 - C_3 \cdot \sin C_4(t - T_0) & (4)
\end{aligned}
$$

where (only in the case of $e = 0$) the parameter $v$ is linearly dependent on time, and $\omega$ may be set to any fixed value, e.g., to $\omega = 90°$. The parameter $C_4 = 2\pi/P$, where $P$ is the initial period and $T_m$ is the weighted mean value of the moments of observations. The parameter $C_4$ is finally determined using differential corrections. The initial epoch $T_0$ then corresponds to the time of a superior conjunction. Then, the best-fit parameters and their errors are determined (see [39] for details).

The parameters of the orbital motion allow "elliptic expansions", which are described in detail in[5]. The non-linear least squares approximation was realized by splitting the parameters of Equation (1) into two "linear" parameters $(\gamma, K)$ and two "non-linear" ones $(e, \omega)$. These latter parameters were determined to minimize the weighted sum of the residuals.

## 3. Results

### 3.1. Known Binary Systems

#### 3.1.1. $\nu$ Gem

$\nu$ Geminorum (BS 2343, [41]) has been known as a multiple system since long ago. History of its studies was recently reviewed in [42], which includes focus on the inner binary and the distant tertiary component. Following the first measurement of the inner binary orbital period (53.73 days) and the conclusion on the absence of phase-locked V/R variations in the double-peaked H$\alpha$ line profile [23], it was assumed that the Be star is the tertiary component, while the inner pair consists of two B-type stars of nearly equal masses.

However, the latest long-baseline interferometry results analysed by Gardner et al. [43] revealed an inconsistency between the astrometric and the mentioned spectroscopic orbital solutions.

We have been monitoring $\nu$ Gem since 2014 at TCO, and our data always show double-peaked H$\alpha$ line profiles with small V/R variations. At the same time, earlier observations published in [23], as well as those found in the BeSS database, the Ondřejov Observatory archive, and the Ritter Observatory archive (some of the latter were taken by one of us, A.M.), show a much larger range of the V/R ratios (see left panel of Figure 1). In addition, the H$\alpha$ line profile in three spectra taken at the Ritter Observatory in November 2001 had four emission peaks. In 2002 a triple-peaked structure was observed at Ritter (see also [23]). It returned to a double-peaked profile in the beginning of 2003, but large V/R variations remained until mid-2006, which was the time period reported in [23]. This behavior has resulted in detecting no regular V/R variations.

The TCO and KO spectra showed a V/R amplitude of $\sim$0.2 and a temporal trend toward smaller V/R. In addition, the V/R ratio showed a strictly periodic phase-locked behavior, which became especially obvious after subtracting a linear trend. Therefore, the Be star takes part in the 53.78-day orbital period and cannot be identified with the tertiary component of the $\nu$ Gem system.

The de-trended V/R variations folded with this period are shown in the bottom panel of Figure 2. The phase curve shows two repeating segments of the sinusoid for a better viewing of the periodic character. Regular variations detected in the properties of all other objects studied in this paper are shown in the same way.

Our long wavelength-range spectra also allowed us to independently derive the binary orbit by cross-correlation using the *rvsao* task in IRAF. Our orbital solution rests on a spectral range between 4450 Å and 4555 Å, where two stronger lines (He I 4471 Å and Mg II 4482 Å) and a few weaker Fe II lines are present (see upper part of the left panel of Figure 1). The RVs of the individual spectra were derived in comparison with a theoretical spectrum for $T_{eff}$ = 14,000 K, log g = 4.0, and $v \sin i$ = 150 km s$^{-1}$, which was calculated using the radiation transfer code *SPECTRUM* [44] and is very close to the observed one. The results are shown in Table 2 and the top panel of Figure 2. We note that the sharp central depression components of the Balmer lines profiles show virtually very small positional variations that do not follow the orbital period.

We note that folding our RV data with half period suggest that the orbit may be slightly elliptical, but the eccentricity is very small. The latter has been found in the abovementioned recent orbital determinations in [42,43], but the measurement uncertainties in our cross-correlated RVs ($\sim$3–5 km s$^{-1}$) hamper detecting the eccentricity.

**Table 2.** Orbital elements derived for $\nu$ Gem.

| Element | [23] | [42] | This Work, e = 0 | This Work, ell |
|---|---|---|---|---|
| P (days) | 53.731 ± 0.017 | 53.761 ± 0.003 | 53.770 ± 0.004 | – |
| T0 (HJD) | 2,451,004.7 ± 4.2 | 2,451,006.5 ± 0.9 | 2,458,417.97 ± 0.05 | – |
| $e$ | 0.11 ± 0.05 | 0.077 ± 0.008 | 0 | 0.036 ± 0.008 |
| $\omega$ (degrees) | 315 ± 29 | 148.7 ± 6.0 | – | 270.0 ± 0.4 |
| $\gamma$ (km s$^{-1}$) | 34.5 ± 1.1 | – | 31.51 ± 0.17 | 31.51 ± 0.16 |
| $K_1$ (km s$^{-1}$) | 38 ± 8 | 48.4 ± 1.4 | 34.54 ± 0.27 | 34.68 ± 0.25 |
| f(m), M$_\odot$ | 0.31 ± 0.16 | 0.64 ± 0.06 | 0.229 ± 0.005 | – |
| N | 45 | 145 | 261 | 261 |

Parameters listed are as follows (line number) : 1—orbital period, 2—periastron epoch for the elliptical orbit and time of a superior conjunction (at $\gamma$ RV) epoch for the circular orbit, 3—eccentricity, 4—argument of the periastron, 5—systemic velocity, 6—semi-amplitude of the RV variation in the visible component, 7—mass function, and 8—number of spectra used in the orbit calculation. The solutions from [23] and [42] (solution C2 from their Table 5) are based on RVs from the He I 6678 Å lines. The TCO and KO solutions based on cross-correlation in a region of 4450–4555 Å (see text) are shown in the last two columns. The elliptical solution shown in the last column has the same period and T0 epoch as those derived in our circular orbit solution. The epochs of T0 from the literature correspond to a minimal RV, while that from our solutions corresponds to a superior conjunction of the Be star.

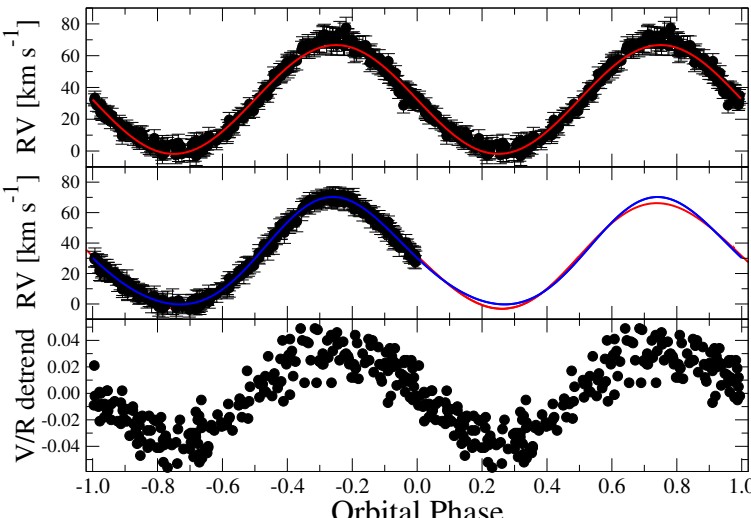

**Figure 2. Top panel:** RV variations in the absorption lines in a region 4450–4555 Å folded with the orbital period. Red line shows our circular solution. **Middle panel:** RV variations in the absorption lines in the spectrum of $\nu$ Gem (same as shown in Figure 1) and best fits by trigonometric polynomials of the first (TP1, red line) and second (TP2, blue line) order. The r.m.s. deviation of the elliptical solution from the TP1 fit is only 0.035 km s$^{-1}$, which is $\approx$5 times smaller than the accuracy estimate of the TP2 approximation. These TP2 $\pm 2\sigma$ curves virtually coincide with each other, but significantly differ from the TP1 (circular orbit) approximation. The second order is statistically significant. The second orbital cycle is shown without the data for an easier inspection of the fits difference. **Bottom panel:** V/R variations in the H$\alpha$ line profile from the TCO and KO data, de-trended with a quadratic function and folded with the same period.

To test this possible small eccentricity, we computed approximations using trigonometric polynomials of the first (TP1) and second (TP2) orders The corresponding False Alarm Probability (FAP) for the statistical significance of TP2 with respect to TP1, calculated from the Analysis of Variances (ANOVA, [39,45]), was FAP = $1.7 \cdot 10^{-5}$. This is due to a 9.3% loss of variance when including the double frequency term in the TP models.

The eccentric model has two more parameters, compared to a circular orbit, i.e., the eccentricity $e$ and the argument of periastron $\omega$ [46]. The corresponding FAP = $2.7 \cdot 10^{-6}$. This means that the TP2 approximation is slightly worse than that found for the elliptic case using Equation (1). This is apparently due to a better fit with an elliptic solution, which contains further harmonics absent in TP2 (see middle panel of Figure 2). The resulting eccentricity is very small and consistent with the previous solutions shown in Table 2.

### 3.1.2. $\kappa$ Dra

$\kappa$ Draconis (BS 4787) is one of the northernmost bright Be stars ($V \sim 3.9$ mag, Declination $\sim 70°$). It was recognized as a binary by Juza et al. [27], who used both photographic and early CCD spectra taken over nearly a century and determined an orbital period of $61.555 \pm 0.003$ days and an RV amplitude of 6–7 km s$^{-1}$. Saad et al. [22,47] used the same set of CCD data with a spectral resolving power $R = 10,000 - 20,000$, taken in 1994–2003, which confirmed the orbital period and found that the V/R ratio of the H$\alpha$ line profile (see Figure 3) showed phase-locked variations with the orbital period. Klement et al. [48] derived the same orbital period using more recent and higher-resolution ($R = 10,000 - 65,000$) spectra but have not studied the V/R variations. In all these studies, RVs of individual spectral lines (e.g., Si II 6347 Å H$\alpha$, He I 4026 Å) were used to determine the orbital period.

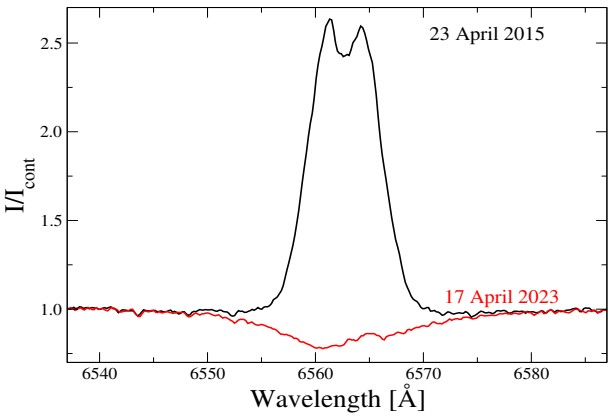

**Figure 3.** Examples of the Hα line profiles of κ Dra at a high emission phase (black line) and a nearly diskless phase (red line). The intensity is normalized to the local continuum; the wavelength scale is heliocentric. The spectra were not corrected for telluric lines.

Our data confirm the previously determined orbital period and the semi-amplitude of the RV variations (see Table 3) both derived using cross-correlation in a spectral region between 4370 and 4500 Å, similar to that used for ν Gem (see top left panel of Figure 4). In addition, the same template as that for ν Gem was used for the cross-correlation of the κ Dra spectra, as both stars have very similar absorption-line strengths. This allowed us to determine the systemic RV of the κ Dra system as well (see right panel of Figure 4).

We also confirm that the V/R variations in the Hα line intensity peaks are phase-locked with the orbital period (see bottom right panel of Figure 4). However, the maxima and minima of the V/R phase curve occur at the RV phases of 0.230 and 0.785, respectively. This is a small effect, but it points out to a slight deviation of the motion of the circumstellar material responsible for the V/R variations from a circular orbit around the Be star.

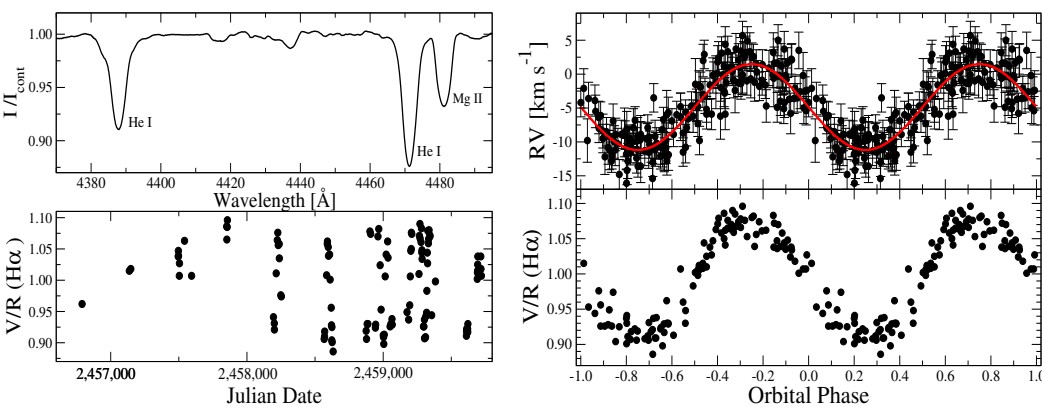

**Figure 4. Left panel**: Top—TCO spectrum of κ Dra averaged from over 100 individual spectra with a typical signal-to-noise ratio of ∼200–300 in the continuum. The spectra were shifted to a common wavelength system before averaging to avoid broadening due to the RV variations. Most of the weak unmarked absorption lines are Fe II lines. Bottom—Temporal behavior of the V/R variations based on TCO and KO data. **Right panel**: Top—RV variations in absorption lines of κ Dra in a region 4360–4500 Å shown in the top left panel folded with the orbital period. Red line shows the best fit to the RV data. Bottom—V/R variations in the Hα line profile from the TCO and KO data folded with the same period. The zero-point of both phase curves is the JD0 parameter listed in Table 3.

**Table 3.** Orbital elements derived for $\kappa$ Dra.

| Element | [47] | [48] | This Work |
|---|---|---|---|
| P (days) | 61.55 $\pm$ 0.02 | 61.5496 $\pm$ 0.0058 | 61.55 $\pm$ 0.04 |
| T0 (HJD) | 2,495,80.22 $\pm$ 0.59 | 2,449,580.4 $\pm$ 1.3 | 2,459,074.45 $\pm$ 0.43 |
| $\gamma$ (km s$^{-1}$) | – | – | $-4.86 \pm 0.18$ |
| $K_1$ (km s$^{-1}$) | 6.81 $\pm$ 0.24 | 6.90 $\pm$ 0.15 | 6.33 $\pm$ 0.25 |
| f(m), M$_\odot$ | 0.0020 $\pm$ 0.0002 | 0.0021 $\pm$ 0.001 | 0.0016 $\pm$ 0.0002 |
| N | 45 | 145 | 190 |

Parameters listed are as follows (line number) : 1—orbital period, 2—periastron epoch for the elliptical orbit and average RV epoch for the circular orbit, 3—systemic velocity, 4—semi-amplitude of the RV variation in the visible component, 5—mass function, and 6—number of spectra used in the orbit calculation. The solutions from [23] and [42] are based on RVs from the He I 6678 Å lines. The TCO and KO solution is based on cross-correlation in a region of 4450–4555 Å (see text).

The bisector RV of the H$\alpha$ line follows the orbital period as well (not shown). Although we took nearly 200 spectra of $\kappa$ Dra, not all of them were used for the V/R variation analysis because the emission-line spectrum has been steadily weakening with time, as was mentioned in [48]. In particular, the H$\alpha$ line became too weak already in 2019 to accurately measure this ratio. Currently (as of May 2023, see Figure 3), the emission component of the H$\alpha$ line is barely detectable.

### 3.1.3. V2119 Cyg

A hot secondary component of V2119 Cyg (BS 7807) shows up through variable RVs in seven UV spectra observed by the IUE and HST [14,49]. These data were used in [49] to derive a preliminary orbital period of 60.29 $\pm$ 0.01 days, which was corrected in [28] to 63.146 $\pm$ 0.003 days by adding astrometric measurements of both components resolved by interferometry.

We took 46 spectra of the system at TCO in 2021–2022 and measured the V/R ratio of the H$\alpha$ line profile, whose examples are shown in top left panel of Figure 5. These measurements were appended with those from the BeSS data (42 spectra taken between 1997 and 2021). The data show a weak linear trend toward smaller V/R values with time (see top right panel of Figure 5). The trend was removed from the entire data collection, whose Fourier power spectrum was calculated. The latter shows the strongest peak at a period of 63.18 days (see bottom left panel of Figure 5). The de-trended V/R measurements were fitted with a sinusoidal function. The best-fit shown in the bottom right panel of Figure 5 was found for a period of 63.16 $\pm$ 0.03 days and a semi-amplitude of 0.019 $\pm$ 0.002. This period shows up even in the original V/R data, but the trend removal lowers the scatter in the data points. The epoch of the maximum V/R coincides with that derived in [28] within the error of the period.

A cross-correlation of various regions in the TCO spectra, which contain at least several absorption lines (e.g., He I and Fe II between 4350 and 4750 Å), with templates based on either one of our spectra of V2119 Cyg or theoretical spectra convolved with the TCO spectral resolution, was also performed. A Fourier spectrum of the resulting RVs shows the highest peak at a very similar period, but a scatter of the RVs folded with the period is large, probably due to a high rotation rate of the Be star (v sin $i \sim 300$ km s$^{-1}$). We can only estimate a semi-amplitude of the RV variations to be $\sim$15 km s$^{-1}$.

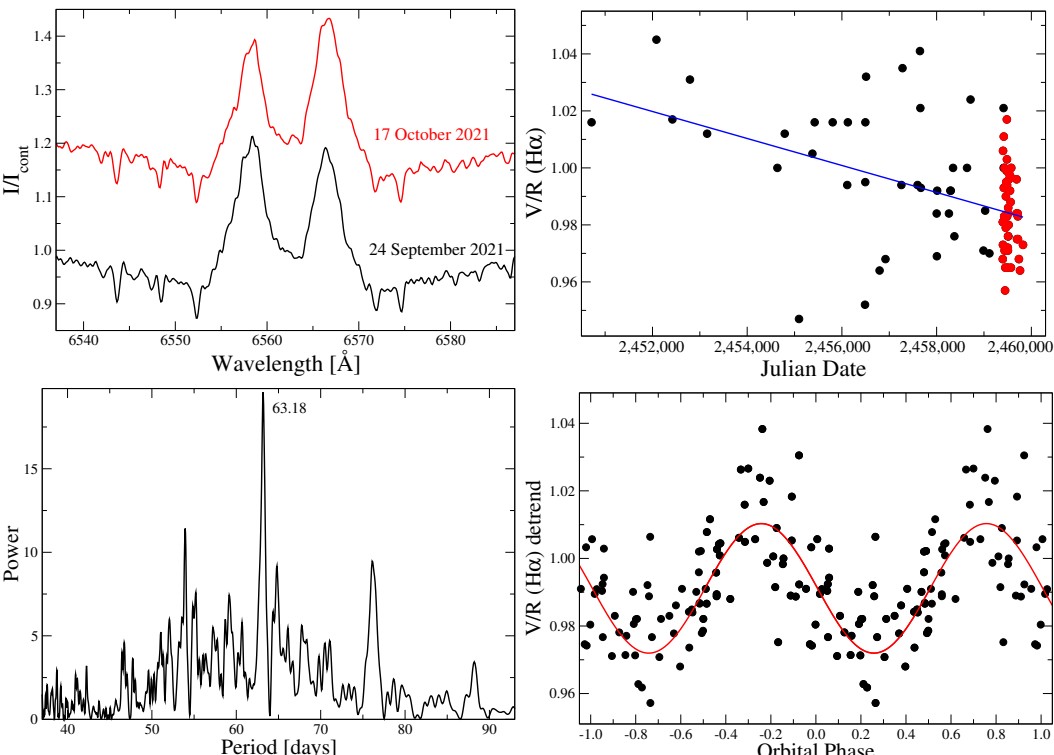

**Figure 5. Left panel**: Top—Examples of the Hα line profiles of V2119 Cyg with the V/R ratio larger than 1 (black line) and less than 1 (red line). The intensity is normalized to the local continuum; the wavelength scale is heliocentric. The spectra were not corrected for telluric lines. Bottom – Fourier power spectrum of the Hα line V/R variations in V2119 Cyg. A linear temporal trend was removed from the measurements. **Right panel**: Top—Temporal behavior of the V/R variations in the Hα line profile in the TCO (red circles) and BeSS (black circles) spectra of V2119 Cyg. The linear trend is shown by the solid blue line. Bottom—De-trended Hα V/R variations folded with an orbital period of 63.16 days. Red line represents a sinusoidal fit to the data. Zero-point of the phase curve corresponds to the epoch of a superior conjunction of the Be star.

### 3.1.4. ε Cap

ε Capricorni (BS 8260) is a bright Be-shell star ($V = 4.5$ mag), whose regular RV variations in the Hδ line with a period of 128.5 days were reported in [23]. Our observations taken between 2017 and 2022 revealed regular variations in both the RVs in a region 4375–4550 Å, measured by cross-correlation and in the V/R ratio of the Hα line peak intensities. Examples of the Hα profiles of ε Cap are shown in left panel of Figure 6.

Additionally, we measured RVs of the absorption line cores of the four Balmer lines (Hα, Hβ, Hγ, and Hδ). All the these lines show very similar RV variations, with a period of 128.98 days. We use the Hγ line RV data to derive the orbital elements (see examples of this line profiles in right panel of Figure 6). The data points scatter on the Hγ RV phase curve is smaller than that on the cross-correlation curve. This difference is probably due to more accurate measurements of the narrow Balmer line cores compared to weaker and broader lines used for the cross-correlation. The orbital periods in both our solutions are within 3σ uncertainty from one another. The Hδ solution from [23] is very close to our Hγ solution. Therefore, we confirm the previously published orbital period and show our results in Table 4 and in Figure 7.

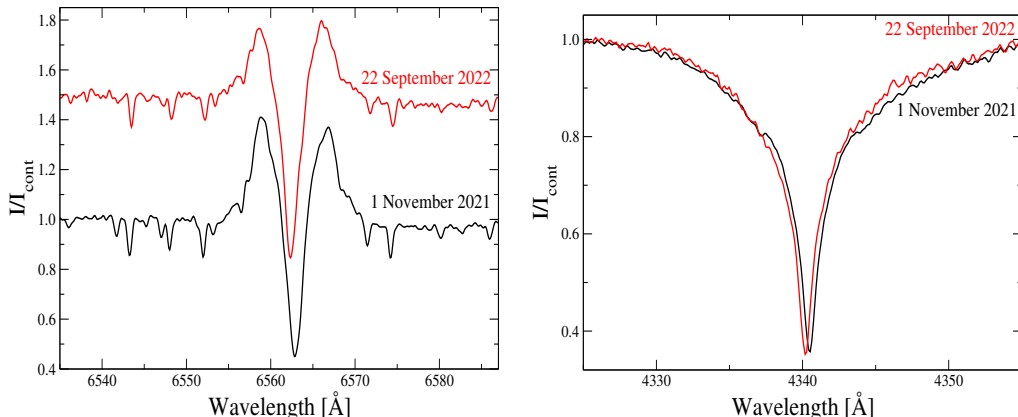

**Figure 6. Left panel**: Examples of the H$\alpha$ line profiles of $\epsilon$ Cap with the V/R ratio larger than 1 (black line) and less than 1 (red line). The spectrum taken on 1 November 2021 corresponds to the orbital phase 0.715, while the spectrum taken on 22 September 2022 corresponds to the orbital phase 0.236. The intensity is normalized to the local continuum; the wavelength scale is heliocentric. The spectra were not corrected for telluric lines. **Right panel**: Examples of the H$\gamma$ line profiles of $\epsilon$ Cap for the same dates as those for the H$\alpha$ lines.

All three T0 epochs listed in Table 4 are shifted with respect to each other. In particular, the shift between our H$\gamma$ solution and that from [23] is 25 days, while our two solutions are shifted by 8 days. The latter offset might be due to a relatively large RV scatter from the best-fitting solutions and the difference in the derived periods. However, the offset between our H$\gamma$ solution and the H$\delta$ solution from [23] might have a different nature and needs further investigation.

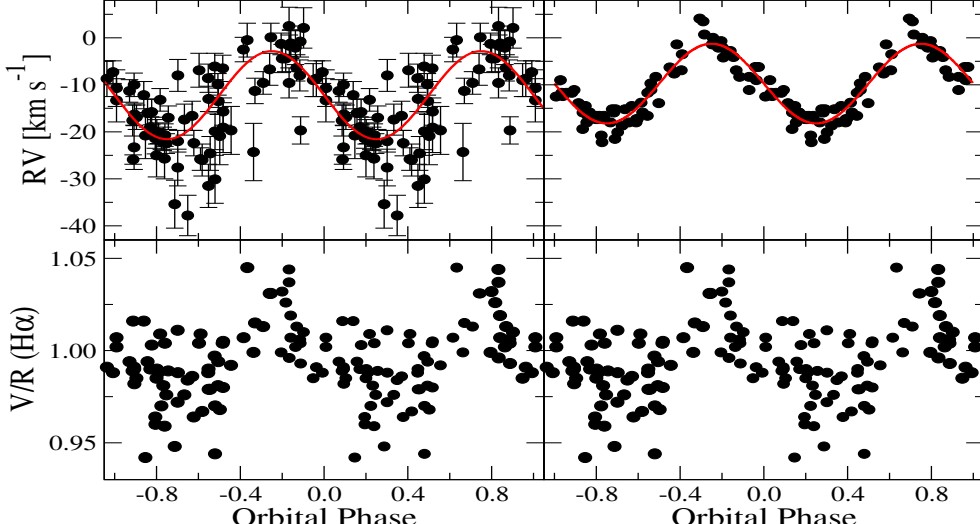

**Figure 7. Left panel**: Top—heliocentric RV variations in absorption lines in a region 4375–4550 Å in the spectra of $\epsilon$ Cap folded with an orbital period of 129.98 days. Bottom—V/R variations in the H$\alpha$ line profile from the TCO data folded with the same period. **Right panel**: Top – RV variations in the H$\gamma$ absorption core in the spectra of $\epsilon$ Cap folded with the orbital period. Bottom—same as in the bottom part of the left panel.

**Table 4.** Orbital elements derived for $\epsilon$ Cap.

| Element | [23] | This Work (cc) | This Work (H$\gamma$) |
|---|---|---|---|
| P (days) | 128.3 ± 0.2 | 130.03 ± 0.57 | 128.98 ± 0.24 |
| T0 (HJD) | 2,451,061.2 ± 0.6 | 2,459,033.4 ± 2.3 | 2,459,041.34 ± 0.98 |
| $\gamma$ (km s$^{-1}$) | −9.2 ± 0.2 | −12.2 ± 0.9 | −9.76 ± 0.27 |
| K$_1$ (km s$^{-1}$) | 8.8 ± 0.3 | 9.36 ± 0.99 | 8.40 ± 0.36 |
| f(m), M$_\odot$ | 0.009 ± 0.001 | 0.011 ± 0.004 | 0.0079 ± 0.001 |
| N | ∼200 | 76 | 73 |

Parameters listed are the same as those in Table 3. The solution from [23] is based on RVs from the H$\delta$ 4101 Å line. The epoch of a superior conjunction for the data from [23] was calculated from the epoch of a maximum RV listed by these authors by adding a quarter of the orbital period. The TCO and KO solution based on cross-correlation in a region of 4375–4550 Å is shown in column 3, while the one based on the H$\gamma$ line RVs is shown in column 4.

### 3.1.5. 60 Cyg

Next, 60 Cygni (BS 8053) was suspected of binarity nearly a century ago, but the first orbit was only presented by Koubský et al. [29] in 2000 based on RV variations in the H$\alpha$ and He I 4471 Å lines. The initial orbit was suggested to be circular with a period of 146.6 ± 0.6 days, while Klement et al. [28], in combining the same 47 data points from [29] with an astrometric orbital solution, came up with an eccentric ($e = 0.2$) orbit and a slightly different orbital period (147.68 ± 0.03 days).

Our data collection is still small (29 spectra taken in 2021–2023). It does not allow us to derive a good orbit from the RV variations in absorption lines due to a large rotational velocity of the Be star ($\geq$300 km s$^{-1}$). Nevertheless, the V/R variations in the H$\alpha$ line profile follow the 147.68-day orbital period very closely (see Figure 8) and are in sync with the RV data from [28]. At the same time, our data permit no independent assessment of the orbital shape and call for more observations.

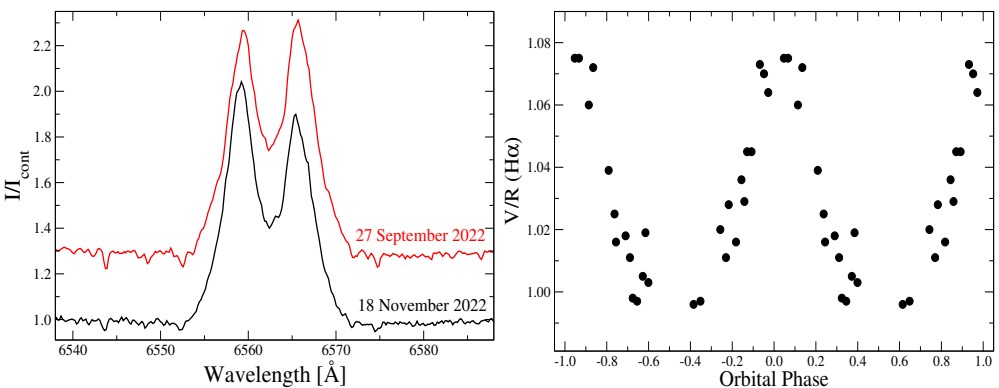

**Figure 8. Left panel**: Examples of the H$\alpha$ line profiles of 60 Cyg with the V/R ratio larger than 1 (black line) and less than 1 (red line). The intensity is normalized to the local continuum; the wavelength scale is heliocentric. The spectra were not corrected for telluric lines. **Right panel**: V/R variations in the H$\alpha$ line profile in the spectrum of 60 Cyg from the TCO data folded with the 147.68-day period and the phase zero-point epoch for a maximal RV from [28].

### 3.1.6. o Pup

o Puppis (BS 3034) has been a candidate binary since Koubský et al. [30] found regular RV variations in some emission lines with a period of 28.903 ± 0.004 days. These authors used 24 spectra taken with different spectrographs, which provided spectral resolving powers of $R = 10,000 − 17,000$, and detected RV variations in the stronger emission peak in a double-peaked He I 6678 Å line profile in anti-phase with those the H$\alpha$ and Paschen lines. These features are suggestive of the presence of an O-type subdwarf (sdO) secondary component. The orbital eccentricity was set to zero. A similar conclusion was reached by

Vanzi et al. [50], who analyzed V/R variations in the He I 6678 Å in 12 spectra and came up with two possible periods of 15.2 and 7.9 days.

We took 64 spectra of the system in 2016–2023 and analyzed the V/R intensity ratio of the emission peaks in a double-peaked H$\beta$ line profile, because the H$\alpha$ line was not always exhibiting a double-peaked profile (Figure 9). Cross-correlation of absorption lines was not attempted because of a very high rotation rate of the Be star ($v \sin i \geq 300$ km s$^{-1}$) and its high T$_{\text{eff}}$. These conditions cause most absorption lines to be very weak and shallow. They are typically very noisy even at signal-to-noise ratios of $\geq$200 in continuum.

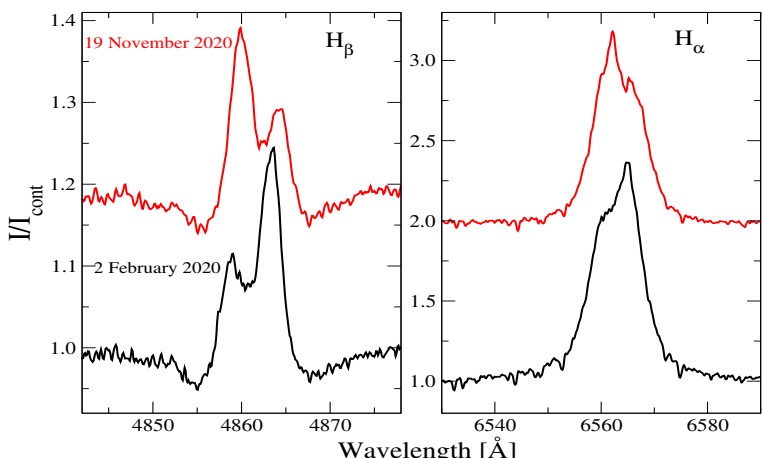

**Figure 9.** Examples of the H$\alpha$ and H$\beta$ line profiles in TCO spectra of o Pup. The intensity is normalized to the local continuum; the wavelength scale is heliocentric. The observing dates shown near the corresponding H$\beta$ profiles are the same for the H$\alpha$ profiles.

The V/R ratio of the H$\beta$ line profile turned out to exhibit variations on at least two time scales. One of them is longer—from season to season (top left panel of Figure 10), while the other one reveals itself after reducing the V/R ratios to the same average level by subtracting seasonal averages from each individual data point. Fitting the de-trended V/R ratio with a TP1 polynomial resulted with a period of $28.17 \pm 0.02$ days (see bottom left panel of Figure 10). These data folded with the latter period are shown in the top right panel of Figure 10. The zero-point of the phase curve corresponds to the time of a superior conjunction of the Be component calculated from the Koubský et al. [30] solution (JD 2,456,048.14). The V/R ratio varies in phase with the RVs of the H$\alpha$ line presented in [30].

We also measured the same emission peak intensity of the He I 6678 Å line as Koubský et al. [30] and analyzed the variations in its RV. The line represents the motion of the secondary component of the binary and varies not only in position but in shape and intensity as well. It is very weak in some orbital phases (especially 0.3–0.4 and 0.8), thus making the position measurements difficult. We managed to measure the peak wavelengths by fitting with a Gaussian using the IRAF task *splot* with a typical accuracy of $\sim$5 km s$^{-1}$. Fourier analysis of these data showed that the strongest peak in the power spectrum corresponds to a period of 28.17 days, the same as that we found for the V/R variations (see right bottom panel of Figure 10). The period was refined through the RV curve fitting with the MCV code (see Section 2.3). Parameters of the system's orbit are presented in Table 5.

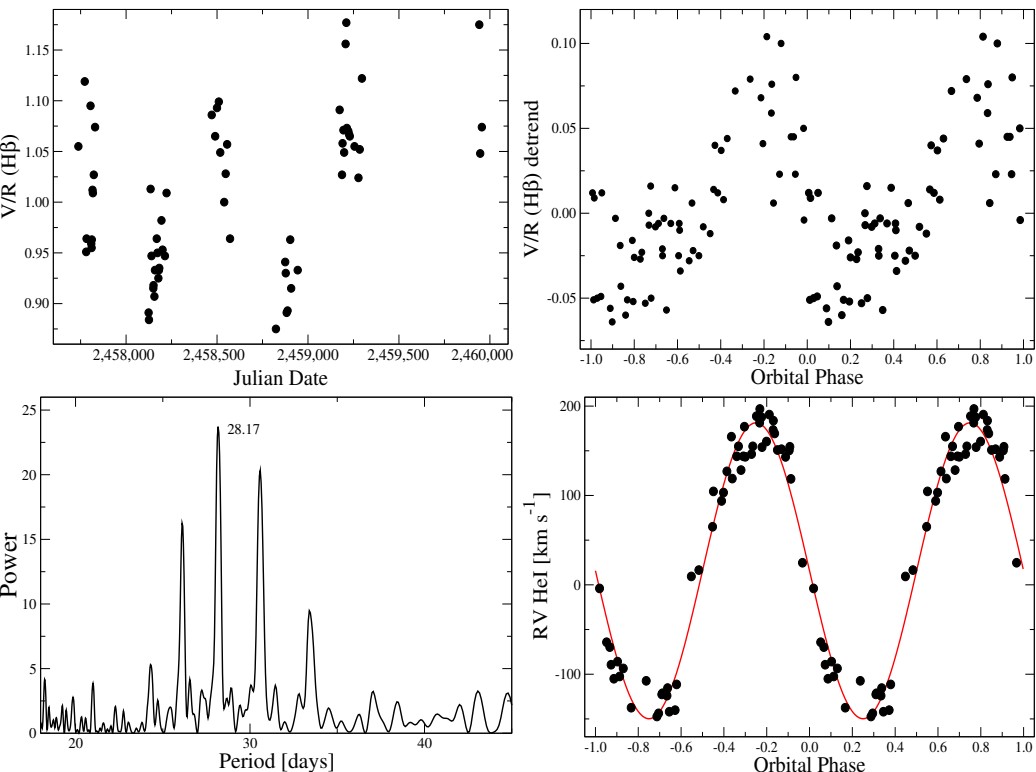

**Figure 10. Left panel**: Top—Temporal behavior of the V/R variations in the Hβ line profile in the TCO spectra of o Pup. Bottom—Periodogram of the He I 6678 Å emission peak RV. **Right panel**: Top—De-trended Hβ V/R variations folded with a period of 28.17 days. The zero-point of the phase curve is T0 = 2,445,721.63, when the primary component was located in a superior conjunction. Bottom—RVs of the He I 6678 Å emission peak folded with a period of 28.17 days (black circles). The zero-point of the phase curve is T0 = 2,445,735.71, when the secondary component was located in a superior conjunction. Red line shows the best fit orbital solution, whose parameters are shown in Table 5.

We did not attempt deriving the bisector RVs of the Hα line, as was performed in [30], because the profiles are variably asymmetric and may not reflect the primary's motion adequately and unambiguously. This task is left for a future study.

**Table 5.** Orbital elements derived for o Pup.

| Element | [30] | This Work |
|---|---|---|
| P (days) | 28.903 ± 0.004 | 28.162 ± 0.007 |
| T0 (HJD) | 2,456,012.93 ± 0.04 | 2,457,735.71 ± 0.13 |
| $\gamma$ (km s$^{-1}$) | 25.0 ± 5.1 | 15 ± 3 |
| $K_1$ (km s$^{-1}$) | 10.3 ± 9.6 | − |
| $K_2$ (km s$^{-1}$) | 159.7 ± 11.7 | 165 ± 4 |
| $f(m_2)$, M$_\odot$ | 12.2 ± 2.5 | 13.1 ± 0.9 |
| N | 16 | 53 |

Parameters listed are as follows (line number) : 1—orbital period, 2—a maximum RV epoch is listed for the solution from [30] and a superior conjunction epoch of the secondary component is listed for our solution, 3—systemic velocity, 4,5—semi-amplitude of the RV variation in the system components ($K_1$—Be star, $K_2$—sdO), 6—mass function, and 7—number of spectra used in the orbit calculation.

### 3.2. Suspected and Unknown Binaries

The rest of our sample contains Be stars which have either not been identified as binaries or their binarity was suspected but not firmly confirmed. Examples of their Hα profiles are shown in Figure 11.

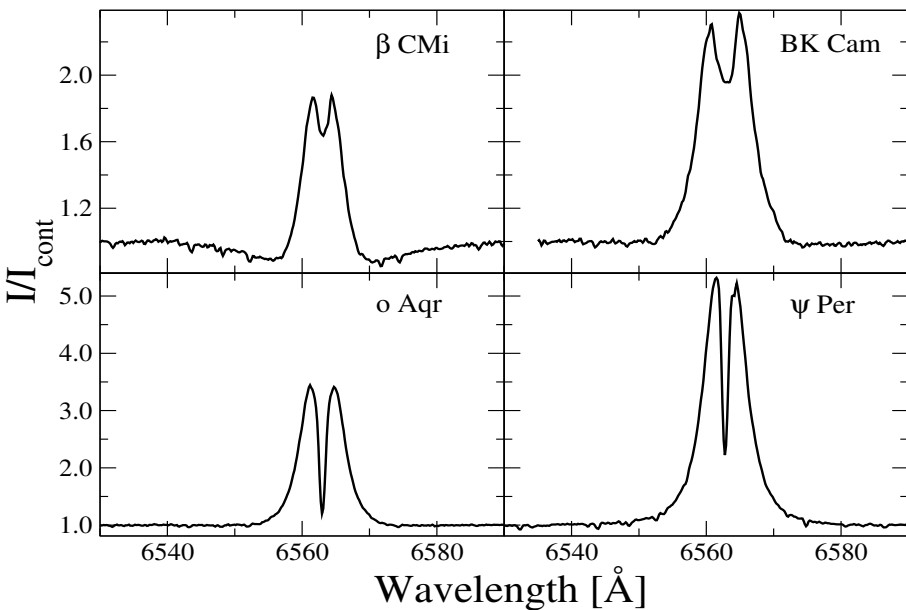

**Figure 11.** Examples of the Hα line profiles in TCO spectra of o Aqr (**bottom left**), ψ Per (**bottom right**), β CMi (**top left**) , and BK Cam (**top right**). The intensity is normalized to the local continuum; the wavelength scale is heliocentric.

### 3.2.1. BK Cam

BK Cam (BS 985) has been observed rather extensively since the beginning of the 20th century. McLaughlin [51] reviewed photographic spectra taken before the 1960s and mentioned that regular V/R variations in the hydrogen line profile were noticed as early as in 1903. He also reported that the V/R variations observed between 1916 and 1931 were regular with a period of 4.5 years, but later they disappeared.

Wang et al. [14] have not detected the presence of a hot (sdO) secondary in the IUE spectra of BK Cam, as the signal-to-noise ratio in the only two available spectra was very low. Finally, Hutter et al. [34] found a very close star near BK Cam using optical interferometry, but these authors only suggested that it might be a physical component with a very long (decades) orbital period. The discovered star is located at an angular distance of ∼130 mas from the Be star and ∼2.5 mag fainter at λ = 700 nm. At a distance of BK Cam ($246^{+19}_{-14}$ pc, [52]), the angular separation corresponds to a linear distance of ∼30 au.

Although the Be star is rather bright, no systematic spectroscopic observations of it has been published since the 1960s (see above). We took almost 100 spectra at TCO in 2018–2023 and measured V/R ratios in the double-peaked Hα and Hβ lines, and also performed cross-correlation of a spectrum region between 4375 and 4560 Å, whose major features are the He I 4387 and 4471 Å lines and the Mg II 4482 Å line. Periodograms of all these time series show a dominant peak at a period of 115 ± 1 day, but all of them result in phase curves with a rather large scatter of data points. The best periodogram and the phase curve was found for the V/R variations in the Hβ line profile, which are shown in Figure 12.

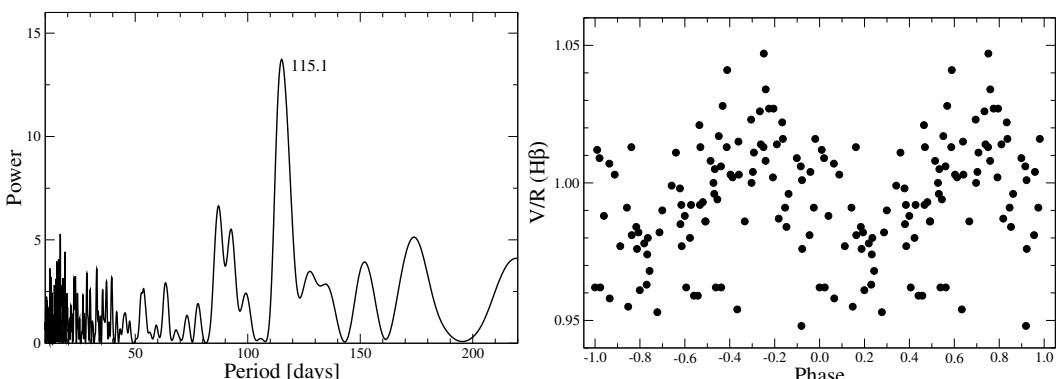

**Figure 12. Left panel**: Fourier power spectrum of the H$\beta$ line V/R variations in BK Cam. **Right panel**: H$\beta$ V/R variations folded with an orbital period of 115.1 days.

### 3.2.2. $\beta$ CMi

$\beta$ Canis Minoris (BS 2845), one of the brightest and closest Be stars, was first suspected in binarity by Struve [53] based on RV measurements on 12 photographic plates with no conclusion on periodicity. A few different orbital periods have been derived from various sets of spectra over the years. In particular, Jarad et al. [54] found a period of 218.5 days from RV variations in non-hydrogen lines in the blue part of the spectrum, Folsom et al. [31] derived a period of 182.83 days from V/R variations in the H$\alpha$ line profile using spectra taken at TCO in 2011–2014, while Dulaney et al. [36] suggested a period of 170.4 days from analysis of RV variations in the H$\alpha$ line wings. The latter result was re-analyzed by Harmanec et al. [55], who have not confirmed the periodicity and suggested that the small RV variations might have been due to rapid H$\alpha$ line profile variations.

Nevertheless, Klement et al. [13] suggested that their earlier result on the disk size (40 stellar radii) corresponds to the 3:2 orbital resonance between the companion orbit for the 170-day period and the disk size. Results from interferometry reviewed in [34] placed a few limits on the angular distance ($\leq$38 mas) and the secondary detection limit ($\Delta$ m = 4.38 mag at $\lambda$ = 692 nm and a distance of 200 mas). Assuming a distance to $\beta$ CMi of 50 pc, a total mass of the system of $\sim$4 M$_\odot$, and nearly a half-year orbital period, leads to an angular separation estimate of $\sim$20 mas, which is smaller than the abovementioned detection limit. Also, a weak central depression in the H$\alpha$ line double-peaked profile implies a relatively large inclination angle of the Be star's rotational axis to the line of sight, making the RV variations small.

Our monitoring of $\beta$ CMi at TCO began in 2011 and is still ongoing. The first results based on the $\sim$200 V/R measurements of the H$\alpha$ line profile taken at TCO, Ritter Observatory, and BeSS database, as well as those presented in [31], showed two strong peaks in the periodogram at 182.83 and 368.23 days. The latter one is present in the spectral window function and is not real. The former peak is much narrower, does not show up in the spectral window, and is still present in the Fourier spectrum calculated for our current data set of over 230 spectra (see left panel of Figure 13). Although the data scatter on the phase curve (right panel of Figure 13) is large, the regularity is clearly seen. The central depression of the H$\beta$ line profile shows small ($\pm$5 km s$^{-1}$) and irregular RV variations.

Our data on other objects (e.g., $\nu$ Gem and $\kappa$ Dra) have been taken with about the same cadence, but their analysis shows no such peaks in the Fourier power spectrum neither for the V/R nor for the RV variations. Therefore, the search for the secondary component in this system is not over yet.

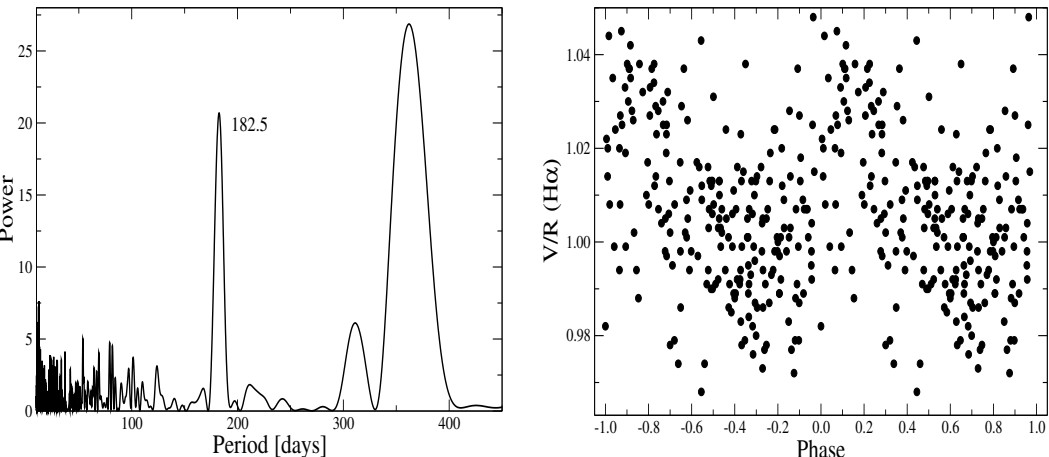

**Figure 13. Left panel**: Fourier power spectrum of the Hα line V/R variations in β CMi. **Right panel**: Hα V/R variations folded with an orbital period of 182.5 days.

### 3.2.3. ψ Per

ψ Persei (BS 1087) received significant attention over the last century, but no clear detection of a secondary component has been reported in the literature. Wang et al. [14] have found no traces of a hot secondary in its UV spectrum. Hutter et al. [34] listed constraints on the angular and brightness detectability of the secondary that are very similar to those mentioned above for β CMi and concluded that ψ Per is a single star.

Klement et al. [13] studied IR excesses of 57 Be stars and found that radio fluxes in ψ Per were consistent with the circumstellar disk truncation at a distance of 100 stellar radii, which can be considered a sign of the presence of a secondary component. However, spectroscopic studies of ψ Per did not mention binarity. Krugov [56] reported that Hα and Hβ emission lines in its spectrum were variable with a period of a few years. Barnsley and Steele [57] also reported variations in the Hα emission between 1995 and 2010, but no conclusion on possible regular changes were made.

We measured V/R and bisector RVs of the Hα line as well as the RVs of the central depressions in the Hβ, Hγ, and Hδ lines. The results for the former show the presence of a 179.6-day period (see left panel of Figure 14), while those for the RVs show no significant peak in the power spectrum at this period. This situation is similar to that found in the data for β CMi, albeit with a noisy V/R phase curve (see right panel of Figure 14).

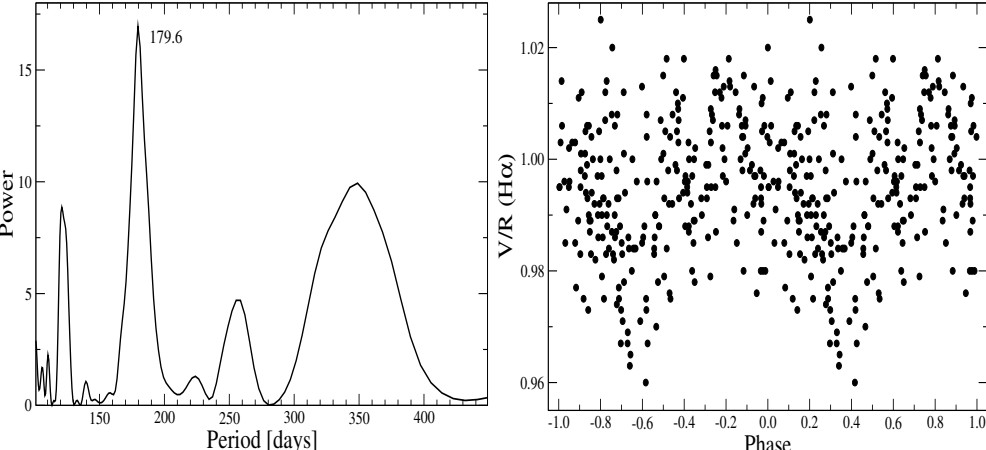

**Figure 14. Left panel**: Fourier power spectrum of the Hα line V/R variations in ψ Per. **Right panel**: Hα V/R variations folded with an orbital period of 179.6 days. Zero-point of the phase curve is chosen arbitrarily.

### 3.2.4. o Aqr

o Aqr (BS 8402) is a Be-shell star with a spectrum similar to that of $\psi$ Per but with weaker emission-line strengths. We have collected over 100 spectra, mostly taken in the 21st century, and took ~160 in 2020–2023 at TCO. Initial results of our study of this collection (with only 40 TCO spectra) were presented in [58].

The V/R ratio in the H$\alpha$ line profile varies from 0.96 to 1.04. The Fourier spectrum of these data shows a dominant period of 72.6 ± 0.5 days in all the datasets that we have collected (TCO, BeSS, and data from [59]). However, RVs of the Balmer line profiles show very little variations, similar to those of $\beta$ CMi, $\psi$ Per, and EW Lac. A periodogram for the V/R variations and a phase curve for the highest confidence period are shown in Figure 15.

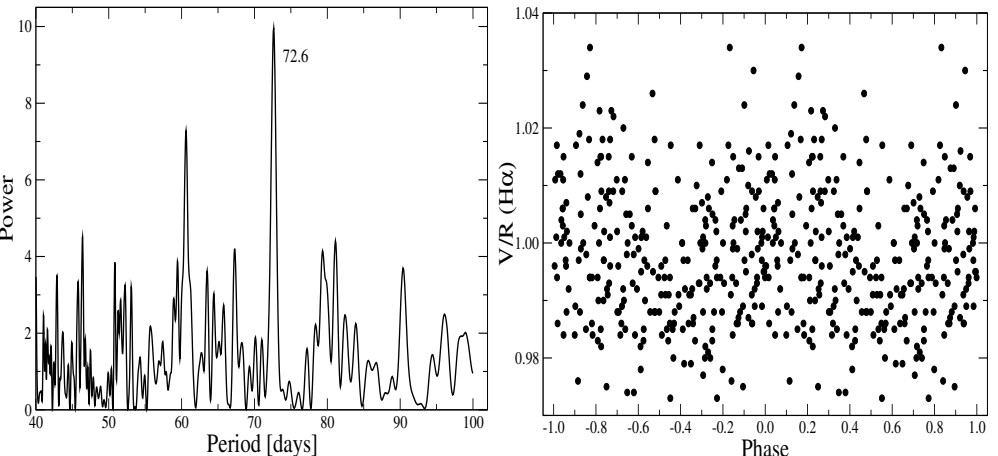

**Figure 15. Left panel**: Fourier power spectrum of the H$\alpha$ line V/R variations in o Aqr. **Right panel**: H$\alpha$ V/R variations folded with an orbital period of 72.6 days. Zero-point of the phase curve is chosen arbitrarily.

### 3.2.5. EW Lac

EW Lac (BS 8731) is a Be-shell star, which exhibited strong variations in the emission-line spectrum that were well-documented by photographic spectroscopy in the 1960s–1980s [60,61] and by CCD spectra in 1992–2002 [23]. All the quoted papers mentioned noticeable V/R variations in double-peaked Balmer emission lines. The emission-line spectrum was the strongest in the 1970s, when the H$\alpha$ EW reached 60 Å [61]. It declined significantly in the 1990s and nearly vanished by 2020 (see left panel of Figure 16).

The V/R ratio of the H$\alpha$ line profile also exhibited dramatic variations in the data reported by Rivinius et al. [23] as well as in our data shown in right panel of Figure 16. Its behavior is similar to that of $\nu$ Gem (Figure 1) with slow and large V/R changes, which settled down in the most recent years (2013–2019) and became reminiscent of those showed by most objects discussed above. In the right panel of Figure 16, we show variations in the H$\alpha$ line blueshifted emission peak intensity that do not always reflect the temporal behavior of the V/R ratio. The two parameters correlate well when the H$\alpha$ line is relatively strong, while this correlation vanishes when the emission-line spectrum becomes weak.

Nevertheless, no dominant period was found in the Fourier power spectrum of the V/R variations during the 6-year period, when the V/R slightly fluctuated around 1. In our analysis, we used H$\alpha$ line measurements from 70 spectra published in [23], 53 spectra taken in 2001–2018 from BeSS, and our data. The V/R ratio was not measured in the spectra with very weak H$\alpha$ lines (2020–2022).

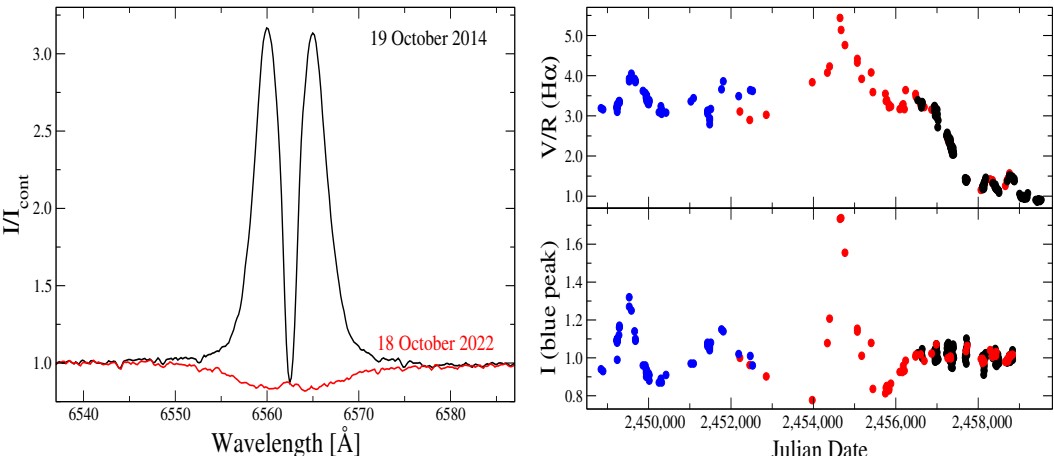

**Figure 16. Left panel:** Sample Hα profiles of EW Lac taken at TCO. The intensity is normalized to the local continuum; the wavelength scale is heliocentric. The observing dates are shown. **Right panel:** Variations of the Hα line blue peak intensity and V/R in CCD spectra of EW Lac. Symbols: black—TCO and KO, red—BeSS, blue—[23]. The blue peak intensity is shown in units of the local continuum.

### 3.2.6. 10 Cas

Finally, 10 Cas (BS 7) is the lowest temperature Be star in our sample (spectral type B9). It has not been closely studied, and only a few spectroscopic observations have been reported in the literature (e.g., [62–64]). The Hα profile is double-peaked (except in the spectrum presented in [63] where it is single-peaked, which might be due to low resolution) and occupies a central part of a broad photospheric absorption line (see top left panel of Figure 17). To the best of our knowledge, no significant variations in the Hα line (which, along with Hβ, is the only emission feature in the object's optical spectrum) have been documented. Therefore, one can assume that the disk of 10 Cas is stable over a long period of time and expect to find long-lived features in its density distribution, which would manifest themselves in the V/R variations.

The V/R ratio in the Hα line profile of 10 Cas exhibits a quadratic trend (red line in the top right panel of Figure 17), which was removed. A Fourier spectrum of the detrended V/R data shows a dominant peak at a period of 40.8 days (see bottom left panel of Figure 17), although the data folded with this period are as noisy as those of o Aqr and ψ Per. Nevertheless, a regular pattern in the V/R variations is noticeable (see bottom right panel of Figure 17).

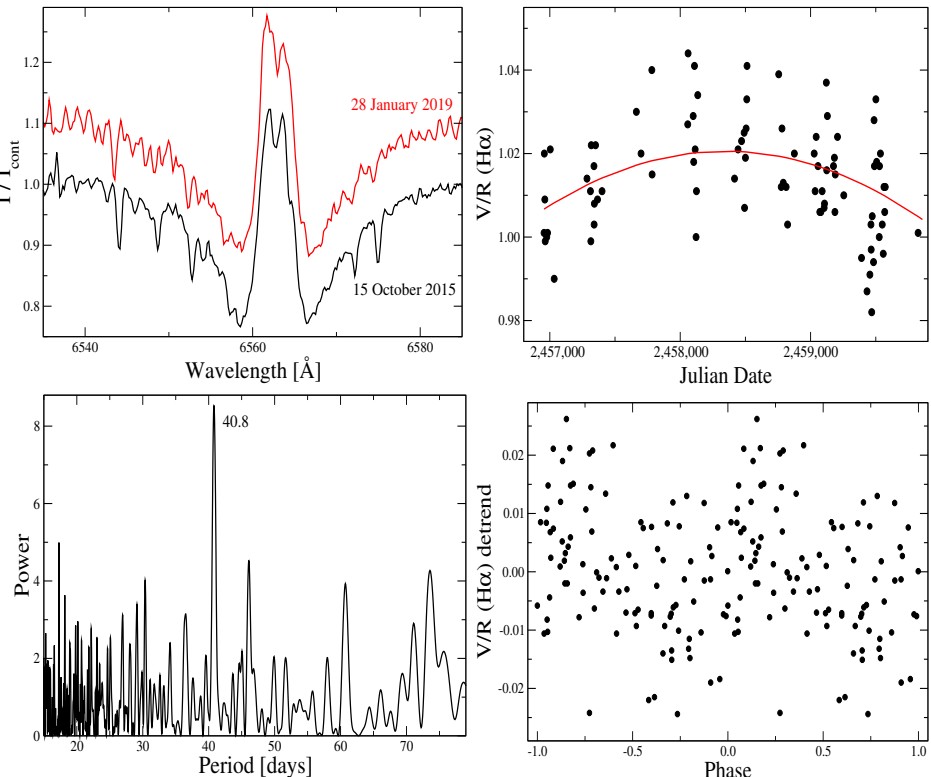

**Figure 17. Left panel**: Top—sample Hα line profiles of 10 Cas taken at TCO with a different V/R ratio. Telluric lines were not taken out. Intensity is normalized to the local continuum, wavelengths are heliocentric. Bottom—Fourier spectrum of the Hα line V/R variations. **Right panel**: Top—V/R variations in the TCO and KO spectra. Red line show a quadratic fit to the data. Bottom—detrended Hα V/R variations folded with an orbital period of 40.8 days.

## 4. Discussion

Our data covered very different time scales of the studied objects. Some of them were observed by us at TCO and KO for nearly 10 years. The collected spectra allowed us to confirm earlier findings for the objects with already known periods (see Table 1), reveal periodicities in others, and obtain more insights into the V/R behavior in general. It was expected from the previous studies that: (1) it takes time for V/R variations to become regular and phase-locked (e.g., [24] for π Aqr); (2) double-peaked emission-line profiles of the Balmer lines (especially those of the Hα line) may turn into more complicated structures (e.g., triple-peaked, see [63] for ν Gem, [65] for π Aqr); (3) the V/R ratio may change significantly and vary randomly (e.g., [61] for EW Lac).

Since the phase-locked V/R variations have been explained as a consequence of the density distribution features, such as density waves [18] or local density enhancements [24], which may be due to tidal interaction between the components of the binary, one can expect that the variations in amplitude would be larger for closer components' masses and shorter orbital periods when the interaction is stronger. At the same time, other factors seem to be affecting this parameter, because its seasonal and longer-term changes are observed as well (see Figures 1 and 10).

Full (from minimum to maximum) amplitudes of the V/R variations in the Hα line profiles of the Be stars with periods determined here and previously (4 Her, 59 Cyg, and π Aqr) are shown in Figure 18. The graph shows a tendency for increasing the V/R amplitudes at shorter periods, which is in agreement with some of the expectations presented above. The obvious outlier is 10 Cas, which has the weakest steady Hα emission. This may be due to an erroneous period or consequences of other factors. The latter may include a large orbital tilt angle, when weak density enhancements may be hidden by the outer parts of the disk (which might be the case for o Aqr), or a small total amount of the circumstellar

gas that, in turn, produces perturbations of a low density that weakly contribute to the line flux.

A misleading position of a star on this graph can also be due to a short-term coverage of its V/R variations which does not fully reveal its true amplitude (e.g., V2119 Cyg). Indeed, our data collection for V2119 Cyg is still relatively small, and the high rotation rate of the Be star that causes the weakness and broadness of the absorption-line spectrum prevented us from deriving a good orbital solution. Nevertheless, we have detected phase-locked V/R variations in the Hα line profile and confirmed the orbital period found in [28].

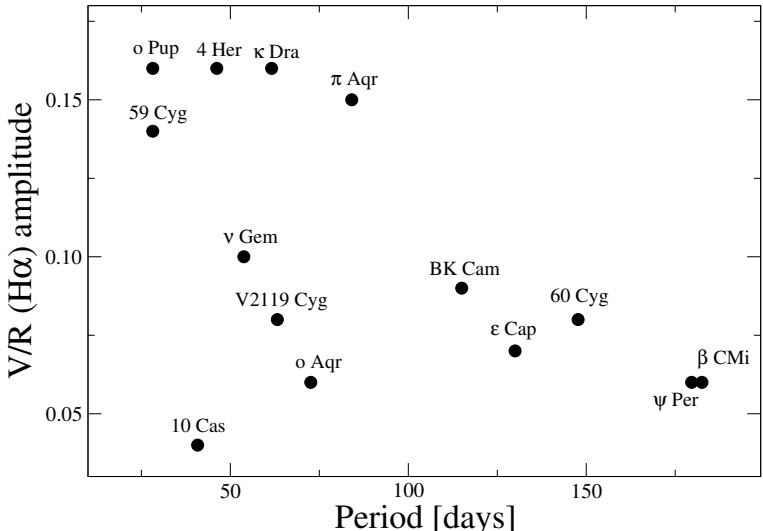

**Figure 18.** Relationship between the Hα line profile V/R amplitude and orbital period for Be binaries and candidates with phase-locked V/R variations. Data for the binaries not studied here were taken from the papers referred to in Section 1.2.

The V/R data for two suspected but yet unconfirmed binaries, $\beta$ CMi and $\psi$ Per, suggest periods of nearly half a year. Such a value can be suspicious, because it may reflect the object's seasonal visibility. However, as was mentioned above (Section 3.2.2), the 182-day period turned out to be the same based on both our earlier analysis of 2014 [31] and that (with twice as many data points) presented here. The V/R phase curve looks noticeably less noisy than those for some confirmed binaries, such as V2119 Cyg and $\epsilon$ Cap. Furthermore, the Fourier spectral window functions, which reveal false periods due to the temporal distribution of observations, show no peaks for both objects. Moreover, recent results presented by Wang et al. [66] revealed a few more SB2 Be+sdO binaries with orbital periods within ±10 days from half a year (HD 113120, HD 137387, and HD 157042).

One of our results, which has not received close attention yet, is strictly in-phased RV and V/R variations observed in all the systems studied here with measurements of both parameters available ($\nu$ Gem, $\kappa$ Dra, $\epsilon$ Cap, o Pup, and 60 Cyg). The relationship between the RV and V/R implies an opposite motion of the Be component and the circumstellar disk material, whose spatial distribution is responsible for the strengths of the emission peaks in a double-peaked line profile. Such a behavior can be explained by a density enhancement that moves in sync with the secondary component of the binary. In particular, this explanation was given to the phased-locked V/R variations in $\pi$ Aqr by Zharikov et al. [24] following the results of the Doppler tomography of a time series of Hα profiles. Although the RV and V/R phase curves of $\pi$ Aqr have not been analyzed together, it is clear that their relationship would follow the same general pattern. We note here that a model for phase-locked V/R variations in binary systems proposed by Panoglou et al. [17] predicts two V/R maxima per orbital cycle, which was observed in no confirmed binary

from our sample. This discrepancy is due to the model assumption of a disk that consists of two spiral arms.

The above discussion suggests that one can expect to see in-phased RV and V/R variations in more Be binaries with moderately strong emission lines. The V/R variations seem to provide good diagnostics for orbital periods, because it is easier to measure this ratio compared to the positions of broad and shallow absorption lines or typically asymmetric emission lines in the spectra of Be stars. Even when the V/R phase curves are very noisy, the Fourier spectrum of the data may show a distinct peak that corresponds to an approximate value of the orbital period. The amplitude of the V/R variations during a stable phase of the emission-line spectrum (when the line strengths vary moderately, such as those of $\nu$ Gem in recent years, see bottom left panel of Figure 1) may give a hint of the expected orbital period.

On the other hand, our results for the Be stars, in whose spectra regular variations in the V/R ratio but not those of the line profile RVs have been detected, may suggest that, although the density enhancements exist in the disk, the star is still single. Such a model to explain long-term V/R variations was proposed by Okazaki [16]. The existence of a density enhancement in the disk of one such object from our sample, EW Lac, in a retrograde motion was suggested by [61]. These objects need further monitoring that may result in a better understanding of both the disk formation and structures therein.

Below, we discuss some implications for the $\nu$ Gem system that follow from our data. Our results unambiguously show that the Be component of this system belongs to the inner binary, as it exhibits strictly periodic variations in the RVs and the V/R peak intensity ratios of the H$\alpha$ profile. The orbital period derived from our data is virtually the same as that found in [42]. The same orbital period also shows up in the RV difference between the H$\alpha$ emission peaks, but the plot is noisier when folded with it. Nevertheless, this parameter is also generally in-phase with the RV and V/R, which points to a certain density distribution around the line formation disk region.

Comparing the orbital solutions from Table 2, one can see that the largest difference between them is in the semi-amplitude of the RV variations. Both previously published ones are based on a single He I 6678 Å absorption line, which is very weak ($\sim$0.98 $I_{cont}$). Our orbital solution described in Section 3.1.1, using RV calculated from cross-correlation, seems to be more reliable, as it is based on the information of $\sim$10 absorption lines. It also allowed us to evaluate the systemic RV ($\gamma$). The latter, derived from all our data (31.51 km s$^{-1}$, see Table 2), is close to the one (29.8 km s$^{-1}$) assumed in [42]. However, the $\gamma$ RV derived from individual observing seasons varies between 30.4 km s$^{-1}$ in 2016/2017 to 34.5 km s$^{-1}$ in 2018/2020, which is most likely due to the influence of the third system component. Nevertheless, taking into account these seasonal shifts leads to virtually no improvement in the best-fit of the RV curve.

Our orbital solution also suggests that the mass of the secondary component of the inner binary (Ab in the notation of [42]) is about 1.5 times smaller compared to that following from the orbital solution derived in [42]. Thus, our result calls for additional observations to reconcile the structure of the system and properties of its components.

## 5. Conclusions

Our study of the V/R variations in double-peaked Balmer emission line profiles in 12 Be stars showed this to be a good method of detecting (or at least suspecting) binarity. We confirmed the presence of the V/R variations phase-locked with the orbital period in all previously known such objects from our sample ($\kappa$ Dra and $\epsilon$ Cap), found such well-defined variations for the first time in $\nu$ Gem, 60 Cyg, and o Pup, refined our previous result for their presence in $\beta$ CMi, and suspected them in BK Cam, $\psi$ Per, o Aqr, and 10 Cas. The only failure of the V/R method was for EW Lac, where no dominant peaks were found in the Fourier spectrum.

Our results doubled the number of known Be binaries with phase-locked V/R variations and suggested that this phenomenon can be observed in spectra of other similar

systems with moderately strong and relatively stable emission-line spectra. Removing a temporal trend from the V/R data results in an easier detection of the periodic component. The latter can manifest itself in the Fourier spectrum as the orbital period in cases of well-defined variations (e.g., ν Gem and κ Dra) or serve as its good approximation if the phase-locked variations are contaminated with other sources of variability (e.g., circumstellar gas inflow/outflow or instability of the tidally indexed density perturbation).

We showed that, in all the confirmed binaries with detected periodic RV and phase-locked V/R variations, both parameters vary in-phase. This implies that the density perturbation/enhancement responsible for the V/R variations moves in sync with the secondary component. The V/R measurements in double-peaked Balmer line profiles (mostly in the Hα and Hβ lines) may help reveal more still elusive binary systems among Be stars and even predict the length of the orbital period by determining the variation amplitude.

An important new result is detecting the phase-locked V/R variations in ν Gem with the orbital period of the inner binary (53.78 days) in this triple system. This will require reconsidering the previously suggested structure of the system, which implied that the Be star was the system's tertiary component on a long-period orbit of nearly two decades [23,42].

**Author Contributions:** Observations, A.S.M., S.D., A.N.A. and P.P.; Data reduction, A.S.M.; Data analysis, A.S.M., R.C., I.L.A., L.L.C., A.L., S.A.K., I.A.G., N.L.V. and S.S.B.; Software A.S.M. and I.L.A.; writing—original draft preparation A.S.M.; writing—review and editing A.S.M., S.D. and I.L.A. All authors have read and agreed to the published version of the manuscript.

**Funding:** This research was funded in part by the Science Committee of the Ministry of Education and Science of the Republic of Kazakhstan (Grant No. AP14972742).

**Data Availability Statement:** Original spectra reported in this study are available on request to the first author via email at a_mirosh@uncg.edu.

**Acknowledgments:** This research has made use of the SIMBAD database, operated at CDS, Strasbourg, France; SAO/NASA ADS; and BeSS database, operated at LESIA, Observatoire de Meudon, France: http://basebe.obspm.fr (accessed on 26 June 2023); data were also provided by the Ritter Observatory Data Archives of the Ritter Astrophysical Research Center, University of Toledo. We thank Daniela Korčáková for sharing spectra of ν Geminorum taken at the 2 m telescope of the Ondřejov Obsevatory (Czechia). The UNCG team acknowledges technical support from Dan Gray (Sidereal Technology company), Joshua Haislip (University of North Carolina Chapel Hill), and Mike Shelton (University of North Carolina Greensboro), as well as the Three College Observatory funding by the College of Arts and Sciences and the Department of Physics and Astronomy at the University of North Carolina Greensboro.

**Conflicts of Interest:** The authors declare no conflicts of interest.

### Abbreviations

The following abbreviations are used in this manuscript: RV—radial velocity, V/R—violet-to-red peak intensity ratio in a double-peaked emission-line profile, R—spectral resolving power, TCO—Three College Observatory, KO—Kernersville Observatory, mas—milliarcsecond, au—Astronomical Unit, BS—Bright Star catalog number.

### Notes

1　http://basebe.obspm.fr/basebe/ (accessed on 26 June 2023).
2　https://www.shelyak.com (accessed on 26 June 2023).
3　https://exoplanetarchive.ipac.caltech.edu/docs/tools.html (accessed on 26 June 2023) (click the Periodogram link).
4　http://uavso.org.ua/mcv/MCV.zip (accessed on 26 June 2023).
5　https://farside.ph.utexas.edu/teaching/celestial/Celestial/node126.html (accessed on 26 June 2023), https://www.astro.uvic.ca/~tatum/celmechs.html (accessed on 26 June 2023).

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
