# Peer review of "Searching for Phase-Locked Variations of the Emission-Line Profiles in Binary Be Stars"

_galaxies, doi:10.3390/galaxies11040083_

Round 1
Reviewer 1 Report
This is a good paper. I recommend it to be accepted, with a few minor corrections:
1. Title : phased-locked - phase-locked
2. The Figure captions: In the figure captions are used a few different labels: V/R, V/R (H-alpha), V/R H-alpha (with Ha and without Ha, with brackets and without brackets). I recommend all the labels to be carefully checked and written in one way. May be something like V/R (Ha) and V/R (Hb), for H-alpha and H-beta, respectively.
3. Please explain how V/R is measured for the cases of H-alpha and H-beta. I guess that for H-alpha, it is something like V/R=(I_V-1)/(I_R-1). Where the continuum level is taken in the case of H-beta (ref. Fig. 7) ?
4. Table 1 – column 6 (Julian day) is missing. Please check all the columns, and may be add column numbers on the second row.
End of the information.
Author Response
Report 1.
This is a good paper. I recommend it to be accepted, with a few minor corrections:
We thank the reviewer for an attentive reading and providing valuable comments. They are all addressed in the corrected version.
- Title: phased-locked - phase-locked − Corrected
- The Figure captions: In the figure captions are used a few different labels: V/R, V/R (H-alpha), V/R H-alpha (with Ha and without Ha, with brackets and without brackets). I recommend all the labels to be carefully checked and written in one way. May be something like V/R (Ha) and V/R (Hb), for H-alpha and H-beta, respectively.
Answer: The notation for the V/R axes was unified.
- Please explain how V/R is measured for the cases of H-alpha and H-beta. I guess that for H-alpha, it is something like V/R=(I_V-1)/(I_R-1). Where the continuum level is taken in the case of H-beta (ref. Fig. 7)?
Answer: The V/R ratio can be defined differently. We measured it by dividing I_V by I_R in the units of the underlying continuum without subtraction of the unity. In the case of such a subtraction, the division errors increase for the peak intensities near the continuum. As seen in Fig. 9 (was Fig. 7 in the original version), the H-beta line in the spectra of omicron Pup is strong enough and was measured in the same way as the H-alpha line. Figure 1 in the corrected version show how the V/R ratio was measured.
- Table 1 – column 6 (Julian day) is missing. Please check all the columns, and may be add column numbers on the second row.
Answer: The range of Julian dates is shown in column 5. The column title was changed from “Date range” to “JD range”. The column numbers are added in the second raw.

Reviewer 2 Report
This paper convincingly establishes the viability of the exploitation of the variability of the intensity ratio between the peaks of double-peaked hydrogen Balmer lines of Be stars to detect their binarity and to constrain their orbital periods. This is a valuable result as it enables a technique of study of the binarity of Be stars for which the classical approach of analysing the radial velocity variations may not be applicable. As a secondary, physically important result, the existence of a correlation between density perturbations in the circumstellar disk and the secondary of binary Be stars is also demonstrated. Given the significance of these results, the paper definitely deserves to be published. However, while overall it is clear and well written, it suffers from some minor issues that need to be addressed before it can be accepted. They are listed below.
1. Page 2, l. 52: “on 340 brightest Be stars”: you probably mean “on the 340 brightest Be stars”. 2. Page 3, l. 125: the acronym BeSS, which is used here for the first time, should be defined. 3. Page 3, l. 148: Usage of “the latter” is inadequate; it does not obviously relate to any element of the previous sentence. 4. Page 4, l. 168: “various humidity”: “various” demands the plural; perhaps use “various humidity levels”. 5. Page 4, l. 169: should it be “profiles” rather than “profile”? 6. Caption of Table 1: “or the range of brightness variations” : no range appears to be given; single values of V are given for all stars. 7. Page 5, l. 209: “circular orbits”: how the orbits are determined could advantageously be described in some detail. 8. Page 5, l. 210: “ellptical” -> “elliptical”. 9. Footnote 3: the link does not work. 10. Footnote 6: the target URL is not found. 11. Figure 1: the orange points are very difficult to distinguish. It may be better to use another colour. 12. Page 8, l. 270: “Drakonis” -> “Draconis”. 13. Page 14, l. 415: “in binarity”: “of binarity” may be better. 14. Page 19, l.551: “noisier” -> “noisy”. 15. Page 20, l. 608” “their components” -> “its components”.
Overall, the English language is good. Some minor errors are identified in my general report. On occasion, phrasing could advantageously be straightened.
Author Response
Report 2.
This paper convincingly establishes the viability of the exploitation of the variability of the intensity ratio between the peaks of double-peaked hydrogen Balmer lines of Be stars to detect their binarity and to constrain their orbital periods. This is a valuable result as it enables a technique of study of the binarity of Be stars for which the classical approach of analysing the radial velocity variations may not be applicable. As a secondary, physically important result, the existence of a correlation between density perturbations in the circumstellar disk and the secondary of binary Be stars is also demonstrated. Given the significance of these results, the paper definitely deserves to be published. However, while overall it is clear and well written, it suffers from some minor issues that need to be addressed before it can be accepted. They are listed below.
We thank the reviewer for a careful reading and suggesting valuable comments. All of the corrections are made in the new version.
- Page 2, l. 52: “on 340 brightest Be stars”: you probably mean “on the 340 brightest Be stars”. – corrected.
- Page 3, l. 125: the acronym BeSS, which is used here for the first time, should be defined. – corrected.
- Page 3, l. 148: Usage of “the latter” is inadequate; it does not obviously relate to any element of the previous sentence. – “the latter” is replaced with “the process”.
- Page 4, l. 168: “various humidity”: “various” demands the plural; perhaps use “various humidity levels”. – corrected to “various humidity levels”.
- Page 4, l. 169: should it be “profiles” rather than “profile”? – changed to “profiles”.
- Caption of Table 1: “or the range of brightness variations” : no range appears to be given; single values of V are given for all stars. – the comment about the range is omitted.
- Page 5, l. 209: “circular orbits”: how the orbits are determined could advantageously be described in some detail. – Formulae describing the orbits were added.
- Page 5, l. 210: “ellptical” -> “elliptical”. – corrected.
- Footnote 3: the link does not work. – the link was replaced with a different one, which leads to a list of tools on the same website.
- Footnote 6: the target URL is not found. - the problem is in LaTeX, which interprets the “~” sign as a an additional space between the words. We’ll report this problem to the technical stuff of the journal.
- Figure 1: the orange points are very difficult to distinguish. It may be better to use another colour. – orange colour for the KO data is replaced with black, the same as that for TCO data, because this is essentially the same data set taken with the same type of spectrograph, same spectral resolving power, and reduced in the exactly same way.
- Page 8, l. 270: “Drakonis” -> “Draconis”. – corrected.
- Page 14, l. 415: “in binarity”: “of binarity” may be better. – corrected.
- Page 19, l.551: “noisier” -> “noisy”. – corrected.
- Page 20, l. 608” “their components” -> “its components”. – corrected.
Reviewer 3 Report
Dear authors,
please find attached my report for your very interesting manuscript "Searching for phase-locked variations of the emission-line profiles in binary Be stars". In my opinion, the manuscripts is well-written and provides an important new view on the emission line profiles of binary Be stars, in particular also on the key system nu Gem.
Please find my report attached below. I sorted my comments to general (more important) and specific (minor) ones.
General:
-------------------------------
1. This paper focuses on V/R variations. What I think is missing in the paper (probably in Sect. 2.2 or 2.3) is a clear description of what V/R variations are, and how they are measured in the first place. Somewhere you mention that a CCF is used but it remains unclear how exactly the V/R variations are defined and measured. A description of this should be added, in particular also with a figure that shows one or two spectra and how the changing V/R variations are measured.
2. In general, for all of the stars: while overall I think it is clear how you handle the data, the details remain unclear. Please, for each of the stars, include the same set of Figures (potentially also in the appendix, if it's too many figures for the main text):
- show H alpha line or V/R variations that are actually visible in the particular star (rather than, for example in the case of nu Gem, a different region of the spectrum)
- in case detrending is applied, please overplot the trend that was removed or show the curves before and after detrending
- add a legend in the figures to indicate which colors correspond to which data points
- if available: add errors to the data points
- for all the figures: make sure that axis labels are correct. For some Figures in the current version, the axis says "VR [km/s]" (e.g., Fig. 3, Fig. 5), which I guess should be "RV" and is quite confusing with "V/R". They should also contain information about which lines were used (i.e., Halpha, Hgamma, ..)
- in all the phase-folded curves, you show the phase from -1 to +1. This might be a bit misleading (i.e., by repeating the same dataset imply more periodicity to the reader than there actually is). If you prefer to stick to this, please mention it explicitly in the text (for example with the first phase-folded curve, and mention it's done for all of them).
- in Fig. 13, you have "I" in the second panel. What exactly is this, and why are you choosing to show it here and not consistently for all stars?
3. Concerning nu Gem:
- firstly, in line 239, you mention that V/R < 1, however, that looks different in Fig. 1. Could you please clarify?
- in the caption of Fig. 1, you mention "orange - KO", however, I cannot see any orange data points. A legend would help.
- can you exclude that such periodic V/R variations could also be induced in a quasi-stationary Halpha line by the motion of two stars in the inner binary?
- please add a physical interpretation of your finding that the Be star is a member of the inner binary, and put it into context, especially in the binary-evolution sketched in the intro?
4. In quite a few stars, you remove a "temporal" trend from the V/R data. Is there a physical interpretation of what could potentially cause this?
Specific comments:
-------------------------------
- Sect. 1.1: quite some literature is discussed here already, but I think in the binary context, also the works by Pols et al. 1991 (A&A 241 419), Shao & Li 2014 (ApJ 796 37), Bodensteiner et al. 2020 (A&A 641 42) and Hastings et al. 2021 (A&A 653 144) would be worth mentioning.
- Sect. 1.2: the sentence "Be stars with strong emission lines due to high disk densities may exhibit V/R variations with much longer periods than orbital ones" remains unclear to me - what would those V/R variations come from (see also comment above)?
- Sect. 1.2: could you please provide more detail on the target and the sample selection. Why do you focus on these 12 stars?
- Sect. 2.1: usage of BeSS spectra: are the BeSS spectra also used to measure RVs? In case they are, how was it ensured that the wavelength calibration was properly performed? (a wrong calibration might lead to the impression of a RV shift). If this was not verified, it should at least be mentioned.
- Sect. 2.2: "but the results turned out to be meaningful only for those with the strongest and narrowest lines" => what is meant by "meaningful"? How do not-meaningful results stand out?
- line 318: "probably due to a high projected RV of the Be star (vsini ..." => instead of RV, do you mean rotational velocity?
- line 350: "follow the orbital period very closely" => please mention which period you are referring to exactly. The one derived above?
- line 566: you mention that the model of Panoglou+ predicts two V/R maxima per orbital cycle. Why is this the case, and what does it imply if you do not measure those?
The overall English is good (please note, I am not native either). I only found a few typos:
- line 210: elliptical
- line 259: "." missing
- line 282: remove "," at the end of the line
- line 337: derived
- line 339: investigation
- line 345: came up
- line 374: are shown in the right panel
- line 427: Deltam= => math mode?
Author Response
Dear authors,
please find attached my report for your very interesting manuscript "Searching for phase-locked variations of the emission-line profiles in binary Be stars". In my opinion, the manuscripts is well-written and provides an important new view on the emission line profiles of binary Be stars, in particular also on the key system nu Gem.
Please find my report attached below. I sorted my comments to general (more important) and specific (minor) ones.
We thank the reviewer for a very attentive reading of our original manuscript. We addressed all the comments and made appropriate corrections in the updated text. We have a bit different opinion on the role of interpretation in the paper but have tried to update according to the comments.
General:
-------------------------------
- This paper focuses on V/R variations. What I think is missing in the paper (probably in Sect. 2.2 or 2.3) is a clear description of what V/R variations are, and how they are measured in the first place. Somewhere you mention that a CCF is used but it remains unclear how exactly the V/R variations are defined and measured. A description of this should be added, in particular also with a figure that shows one or two spectra and how the changing V/R variations are measured.
Answer: We modified Fig. 1 to show how the V and R emission components were measured as well as an explanation of its measurement in the text. The description of what the V/R ratio is can be found in Sect. 1.2.
- In general, for all of the stars: while overall I think it is clear how you handle the data, the details remain unclear. Please, for each of the stars, include the same set of Figures (potentially also in the appendix, if it's too many figures for the main text):
- show H alpha line or V/R variations that are actually visible in the particular star (rather than, for example in the case of nu Gem, a different region of the spectrum)
Answer: The Halpha line profiles and/or its V/R variations are shown for all the discussed stars. A new figure with the profiles for four stars with still questionable binarity was added. Since the stars show different behavior, the sets of figures are not the same, but the Halpha profiles with different V/R are now shown for all the objects with detected regular variations.
- in case detrending is applied, please overplot the trend that was removed or show the curves before and after detrending
Answer: The V/R variations before and after detrending were shown for all the stars where a trend was found.
- add a legend in the figures to indicate which colors correspond to which data points
Answer: This has been done even in the original version, but we checked that this information in the new version is correct.
- if available: add errors to the data points
Answer: If the errors are not present in the Figures, they are either about the size of the data symbols or obstruct the loosely constrained periodic variations. We added notes about the errors in all relevant cases.
- for all the figures: make sure that axis labels are correct. For some Figures in the current version, the axis says "VR [km/s]" (e.g., Fig. 3, Fig. 5), which I guess should be "RV" and is quite confusing with "V/R". They should also contain information about which lines were used (i.e., Halpha, Hgamma, ..)
Answer: We have checked all the axis labels and corrected the confusion. The line IDs (H-alpha, H-beta) have been added to the labels.
- in all the phase-folded curves, you show the phase from -1 to +1. This might be a bit misleading (i.e., by repeating the same dataset imply more periodicity to the reader than there actually is). If you prefer to stick to this, please mention it explicitly in the text (for example with the first phase-folded curve, and mention it's done for all of them).
Answer: This kind of the phase curve representation is quite common in studies of binary systems. It allows to better see the sinusoidal behavior than just one segment. We added a sentence about this to Section 3.1.1, where Figure 1 is described.
- in Fig. 13, you have "I" in the second panel. What exactly is this, and why are you choosing to show it here and not consistently for all stars?
Answer: The meaning of the “I” is explained in the Figure 16 caption. It is shown to illustrate the rather strong variations of the Halpha emission line in this only object, where no clear signs of periodic variations have been found. Essentially, the right panel of Fig. 16 characterizes the long-term variations of the H-alpha line profile. A sentence about the blue-peak intensity behavior is added to the text (Sect. 3.2.5.).
- Concerning nu Gem:
- firstly, in line 239, you mention that V/R < 1, however, that looks different in Fig. 1. Could you please clarify?
Answer: This effect is partially due to the filled circle size in Fig.1. We omit this statement, as it is unimportant for our analysis. It is enough to state that there is a weak trend in the V/R data that was removed before the Fourier analysis.
- in the caption of Fig. 1, you mention "orange - KO", however, I cannot see any orange data points. A legend would help.
Answer: orange colour for the KO data is replaced with black, the same as that for TCO data, because this is essentially the same data set taken with the same type of spectrograph.
- can you exclude that such periodic V/R variations could also be induced in a quasi-stationary Halpha line by the motion of two stars in the inner binary?
Answer: We mentioned that the phase-locked V/R variations have been interpreted as a result of moving density enhancements in the primary’s disk. The only possibility to explain the variations by the stars’ motion is to assume eclipses of a disk’s part by the secondary component, but this would lead to an opposite variability pattern than the observed one.
- please add a physical interpretation of your finding that the Be star is a member of the inner binary, and put it into context, especially in the binary-evolution sketched in the intro?
Answer: This is just an observational fact. Also, the spectra do not directly trace the tertiary component. Without the interferometry data, we wouldn’t even know that the system is triple. We do not approach the background or consequences of the binary evolution. This is out of the scope of this paper. Our goal is to report our observational findings and show that the phase-locked V/R variations are more common that it has been thought.
- In quite a few stars, you remove a "temporal" trend from the V/R data. Is there a physical interpretation of what could potentially cause this?
Answer: The existence of such trends may have to do with the inflow/outflow dynamics in the disk. The explanation of these effects is beyond the scope of our paper.
Specific comments:
-------------------------------
- Sect. 1.1: quite some literature is discussed here already, but I think in the binary context, also the works by Pols et al. 1991 (A&A 241 419), Shao & Li 2014 (ApJ 796 37), Bodensteiner et al. 2020 (A&A 641 42) and Hastings et al. 2021 (A&A 653 144) would be worth mentioning.
Answer: We thank the reviewer for providing these references, but these papers are theoretical, while our paper is observational and its goal to test a certain method of finding binary systems among Be stars. Instead of mentioning these papers, we refer to the paper by Naze et al. (2022, MNRAS, 510, 2286), where all these papers are mentioned and another glimpse of Be binaries is provided.
- Sect. 1.2: the sentence "Be stars with strong emission lines due to high disk densities may exhibit V/R variations with much longer periods than orbital ones" remains unclear to me - what would those V/R variations come from (see also comment above)?
Answer: This is an observational fact that we just mention here, because it puts a limit on our sample. Apparently, this is due to a certain outcome of the interplay between gravity and rotation within the circumstellar disk. This fact has not been clearly explained by theorists, and its explanation is beyond the scope of this paper.
- Sect. 1.2: could you please provide more detail on the target and the sample selection. Why do you focus on these 12 stars?
Answer: This topic is discussed in the first paragraph of the Discussion section. We expanded it and moved it to the end of Section 1.2.
- Sect. 2.1: usage of BeSS spectra: are the BeSS spectra also used to measure RVs? In case they are, how was it ensured that the wavelength calibration was properly performed? (a wrong calibration might lead to the impression of a RV shift). If this was not verified, it should at least be mentioned.
Answer: We have not used BeSS data for the radial velocity measurements. We only used these data for the V/R measurements.
- Sect. 2.2: "but the results turned out to be meaningful only for those with the strongest and narrowest lines" => what is meant by "meaningful"? How do not-meaningful results stand out?
Answer: What we meant by “meaningful” is detection of periodic variations. Non-meaningful simply means chaotic variations.
- line 318: "probably due to a high projected RV of the Be star (vsini ..." => instead of RV, do you mean rotational velocity?
Answer: Yes, we meant rotational velocity. This is corrected to “rotation rate”.
- line 350: "follow the orbital period very closely" => please mention which period you are referring to exactly. The one derived above?
Answer: the value of the orbital period (147.68-day) is explicitly quoted in this sentence now.
- line 566: you mention that the model of Panoglou+ predicts two V/R maxima per orbital cycle. Why is this the case, and what does it imply if you do not measure those?
Answer: The Panoglou et al. model assumes a two-arm density perturbation in the disk. Our results mean that either the perturbation is one-armed or is a local density enhancement, such as deduced by Mon et al. (2013) for EW Lac or by Zharikov et al. (2013) for π Aqr. We added an explanation to the corrected version.
Round 2
Reviewer 3 Report
Dear authors,
thank you for the revised manuscript taking into account the comments I gave. In particular, the updated figures give a much better idea of the V/R variations and how they were assessed.
I only have one very minor textual comment - see below.
Congratulations on this very interesting piece of work.
Sect. 2.2 - "The spectra were normalized to the local continuum and the peak intensities were measured in units of the continuum without subtracting a unity" => without subtracting unity
Author Response
We thank the reviewer again for a careful reading of the updated manuscript and a high evaluation of our work.
Dear authors,
thank you for the revised manuscript taking into account the comments I gave. In particular, the updated figures give a much better idea of the V/R variations and how they were assessed.
I only have one very minor textual comment - see below.
Congratulations on this very interesting piece of work.
Sect. 2.2 - "The spectra were normalized to the local continuum and the peak intensities were measured in units of the continuum without subtracting a unity" => without subtracting unity
Answer: The comment has been taken into account.